# SCF^RMF mediates degradation of the meiosis-specific recombinase DMC1

Wanyue Xu [1], Yue Yu [1], Juli Jing[2], Zhen Wu[1], Xumin Zhang [1], Chenjiang You[1], Hong Ma [3], Gregory P. Copenhaver [4,5], Yan He[2] & Yingxiang Wang [1,6,7] ✉

Meiotic recombination requires the specific RecA homolog DMC1 recombinase to stabilize strand exchange intermediates in most eukaryotes. Normal DMC1 levels are crucial for its function, yet the regulatory mechanisms of DMC1 stability are unknown in any organism. Here, we show that the degradation of *Arabidopsis* DMC1 by the 26S proteasome depends on F-box proteins RMF1/2-mediated ubiquitination. Furthermore, RMF1/2 interact with the Skp1 ortholog ASK1 to form the ubiquitin ligase complex SCF^RMF1/2. Genetic analyses demonstrate that RMF1/2, ASK1 and DMC1 act in the same pathway downstream of SPO11-1 dependent meiotic DNA double strand break formation and that the proper removal of DMC1 is crucial for meiotic crossover formation. Moreover, six DMC1 lysine residues were identified as important for its ubiquitination but not its interaction with RMF1/2. Our results reveal mechanistic insights into how the stability of a key meiotic recombinase that is broadly conserved in eukaryotes is regulated.

In most sexually reproducing eukaryotes meiotic recombination during prophase I is required to ensure subsequent accurate segregation of homologous chromosomes (homologs), and disrupting recombination causes aneuploidy which in turn can result in reduced fertility and genetic disorders in animals and plants[1,2]. In addition, the exchange of genetic material between homologs that results from recombination creates novel combinations of alleles that contribute to phenotypic diversity and adaptability to environmental stress and change.

Meiotic recombination is initiated by the formation of DNA double-strand breaks (DSBs) catalyzed by the highly conserved topoisomerase-like protein SPO11 and several associated proteins[3]. Following DSB formation, the MRN/X (Mre11-Rad50-Nbs1/Xrs2) complex processes the ends of the break to yield 3' single-stranded overhangs[4]. The single-stranded DNA (ssDNA) forms nucleoprotein filaments with the RecA-related recombinases DMC1 and RAD51, which invade homologs to form a recombination intermediate called a D-loop[5]. The first *recA* mutant was isolated in *E. coli* (*Escherichia coli*) and was found to be defective in recombination and DNA damage repair[6]. RecA has DNA binding and ATPase activity as well as the ability to catalyze ATP-dependent hybridization of ssDNA to homologous double-stranded DNA (dsDNA) to form D-loops[7]. Eukaryotes contain multiple RecA family proteins[8], including RAD51 and the closely related meiosis-specific DMC1, which were both first isolated in budding yeast (*Saccharomyces cerevisiae*). Rad51 is required for both mitotic and meiotic recombination[9], while DMC1 is specifically required for meiotic recombination and crossover formation[10]. Yeast Dmc1 is able to bind single and double-stranded DNA, and mediate invasion of ssDNA into dsDNA to promote strand exchange[11]. DMC1 orthologs have been found in animals, protists, and plants[12], with a conserved role in

[1]State Key Laboratory of Genetic Engineering and Ministry of Education Key Laboratory of Biodiversity Sciences and Ecological Engineering, Institute of Plant Biology, School of Life Sciences, Fudan University, Shanghai, China. [2]MOE Key Laboratory of Crop Heterosis and Utilization, National Maize Improvement Center of China, College of Agronomy and Biotechnology, China Agricultural University, Beijing, China. [3]Department of Biology, the Huck Institutes of the Life Sciences, the Pennsylvania State University, University Park, PA, USA. [4]Department of Biology and the Integrative Program for Biological and Genome Sciences, University of North Carolina at Chapel Hill, Chapel Hill, NC, USA. [5]Lineberger Comprehensive Cancer Center, University of North Carolina School of Medicine, Chapel Hill, NC, USA. [6]College of Life Sciences, Guangdong Provincial Key Laboratory of Protein Function and Regulation in Agricultural Organisms, South China Agricultural University, Guangzhou, China. [7]Guangdong Laboratory for Lingnan Modern Agriculture, Guangzhou, China. ✉e-mail: yx_wang@fudan.edu.cn

meiotic recombination. The assembly of Dmc1 on RPA-coated ssDNA is mediated by Mei5-Sae3[13] in budding yeast and Mnd1-Hop2[14] in plants. In yeast, dissociation of Dmc1 from dsDNA relies on the Rdh54/Tid1 ATPase and its paralog Rad54 when in the absence of Rdh54/Tid1[15], which allows the binding of subsequent recombination factors and is a key step for homologous recombination to proceed. However, very little is understood about the mechanisms that regulate DMC1 stability and degradation. Unlike DMC1, the ubiquitination and degradation of RAD51 has been reported to be regulated by the F-box DNA helicase FBH1 in both yeast and mammals[16,17], and the F-box protein FBX5/EMI1 as well as the RING-type E3 RFWD3 in human[18,19]. Recent studies found that human DMC1 can be ubiquitinated in vitro[18]. Similarly, in mouse, inhibition of ubiquitination or proteasomal degradation leads to abnormal DMC1 turn over[20], indicating that DMC1 might also be regulated by ubiquitination and proteasomal degradation. However, the factors that facilitate DMC1's ubiquitination and its impacts on meiosis are not known in any organism.

Ubiquitination is a versatile post-translational protein modification in eukaryotes. Poly-ubiquitination is required for protein degradation by the proteasome in most cases, while mono-ubiquitination is often involved in other process including protein location and activity[21]. Previous studies found that ubiquitination is also important in meiosis[22]. In budding yeast, the Skp, Cullin, F-box (SCF) complex, and Ufd4 are two major E3 ubiquitin ligases that control cohesin-associated Pds5 in regulating meiotic chromosome axis-associated protein degradation[23]. In mammals, RNF212 and HEI10 are putative SUMO and ubiquitin ligases respectively, and have been identified as meiotic recombination regulators[20]. SKP1, a subunit of the SCF complex, is required for the prophase I to metaphase I transition during male meiosis[24], and SKP1 together with the F-box protein FBXO47 targets HORMAD1 for poly-ubiquitination and proteasomal degradation, thus preventing HORMAD1 from recruiting pre-DSB complexes and restricting DSB formation[25]. PSMA8, a mammalian testis-specific α4-like proteasome subunit, is also essential for meiosis[26]. Bre1-mediated histone H2B K123 mono-ubiquitination in yeast and RNF20/RNF40-mediated H2B K120 mono-ubiquitination in mammals participate in meiotic recombination[27]. In plants, seven meiotic E3 ligase subunits have been reported including: the Skp1 homolog *Arabidopsis* Skp1-like1 (ASK1)[28]; the RING-type proteins *Homo sapiens* Enhancer of Invasion 10 (HEI10) and DESYNAPSIS1 (DSNP1)[29,30]; Anaphase-promoting Complex/cyclosome subunit 8 (APC8)[31]; and the F-box proteins MEIOTIC F-BOX (MOF), and ZYGOTENE1 (ZYGO1) in rice (*Oryza sativa*) and Abnormal Chromosome Organization in Zygotene 1 (ACOZ1) in maize (*Zea mays*)[32–34]. Nevertheless, the direct targets under control of the ubiquitin-proteasome system during meiosis are largely uncharacterized.

We used an immunoprecipitation-mass spectrometry (IP-MS) screen to identify factors that regulate the stability and degradation of DMC1. We demonstrate that DMC1 can be ubiquitinated for subsequent degradation in a 26S proteasome-dependent manner. Intriguingly, we show that two functionally redundant meiosis-specific F-box proteins RMF1 and RMF2, interact with ASK1 and DMC1 using distinct domains, and that ASK1-RMF1/2 can directly mediate DMC1's ubiquitination and protein instability. Mutant analysis shows that ASK1-RMF1/2-DMC1 function in the same meiotic recombination pathway. Taken together, these results provide a mechanistic insight into the regulation of DMC1 ubiquitination by an SCF complex and the identified ubiquitin-proteasome machinery in regulation of meiotic recombination appears to be conserved among eukaryotic lineages.

## Results

### DMC1 is ubiquitinated in vivo
To search for regulators of DMC1 stability, we generated DMC1-FLAG transgenic *Arabidopsis* plants and validated the expression of recombinant DMC1 in vivo using western blots and RT-qPCR (Supplementary

Fig. 1a–c). We then used an immunoprecipitation-mass spectrometry (IP-MS) assay to identify 6477 peptides corresponding to 2309 proteins (Supplementary Data 1). 1009 proteins remained after removing background interactions using plants transformed with the corresponding empty vector (Supplementary Data 2). GO term analysis revealed that ubiquitin-proteasome pathway components are significantly enriched among the candidate proteins (Fig. 1a, Supplementary Data 2), suggesting that DMC1 might be ubiquitinated and degraded.

To corroborate this hypothesis, we employed an in vivo ubiquitination assay using transient expression of DMC1-GFP in tobacco (*Nicotiana benthamiana*) leaves. Western blots probed with antibodies against GFP, UBIQUITIN 11 (UBQ11), or DMC1 have a prominent DMC1-GFP band with a smear of signal above the main band that may correspond to ubiquitinated forms (Fig. 1b). We also examined putative DMC1 ubiquitination in *DMC1-FLAG* transgenic *Arabidopsis* plants and plants expressing GFP-FLAG as control. The anti-UBQ11 and anti-DMC1 (specificity verified in Supplementary Fig. 1d) antibodies detect smears above the primary bands in DMC1-FLAG but not in controls (Fig. 1c). These results are consistent with the hypothesis that DMC1 is ubiquitinated in *Arabidopsis*.

To validate these results, we used an independent TUBE2 agarose bead assays system to examine DMC1's ubiquitination. Tandem-repeated ubiquitin-binding Entities (TUBEs) based on ubiquitin-associated (UBA) domains are an efficient tool to specifically isolate and identify ubiquitinated proteins, rather than ubiquitin-related modifications[35]. We used TUBE2 agarose beads to immunoprecipitate ubiquitinated proteins from tobacco and *Arabidopsis* that transiently or stably express DMC1-FLAG, respectively. The anti-UBQ11 antibody detects ubiquitinated proteins in all samples, while anti-FLAG antibody only detects ubiquitinated DMC1 in the experimental lane but not in controls (Fig. 1d, e).

In both experimental systems, in addition to the smeared DMC1 signal, the main DMC1 band was detected, indicating that the main band might represent constitutively mono-ubiquitinated DMC1. To corroborate this hypothesis, we heterologously expressed and purified recombinant SUMO-HIS-DMC1-GFP and SUMO-HIS-DMC1-FLAG in *E. coli*, and used SUMOase to cleave the SUMO-HIS tag to yield DMC1-GFP and DMC1-FLAG. We found that the molecular weight of recombinant DMC1-GFP is smaller in *E. coli* than in tobacco (Supplementary Fig. 1e), and a similar relationship was observed for DMC1-FLAG (Supplementary Fig. 1f), likely due to lack of ubiquitination in *E. coli*. In addition, an anti-UBQ11 antibody can recognize DMC1 expressed in either tobacco or *Arabidopsis*, but not DMC1 from *E. coli* (Supplementary Fig. 1e, f), indicating that DMC1 can be mono-ubiquitinated and poly-ubiquitinated *in planta*. To examine whether mono-ubiquitinated DMC1 is present in meiocytes or induced by somatic expression, we purified recombinant SUMO-HIS-DMC1 from *E. coli* and used SUMOase to cleave the SUMO-HIS to yield untagged DMC1. We examined DMC1 expressed in *E. coli* and endogenous *Arabidopsis* Col-0 DMC1 by western blotting using an anti-DMC1 antibody. DMC1 expressed in *E. coli* has the same molecular weight as the endogenous DMC1 from Col-0 central inflorescences, and endogenous DMC1 cannot be detected in Col-0 peripheral inflorescences and leaves, or in *dmc1-2* central inflorescences (Supplementary Fig. 1g). These results indicate that the mono-ubiquitinated DMC1 is not present at detectable levels in meiocytes, but that mono-ubiquitinated DMC1 may be induced by ectopic expression in somatic cells. Taken together, our results provide strong evidence that DMC1 is poly-ubiquitinated in *Arabidopsis* meiocytes.

### Identification of two functionally redundant meiotic F-box proteins
Previous studies have identified three meiotic E3 ligase components in *Arabidopsis*, including: APC8 of the multi-subunit E3 ubiquitin ligase

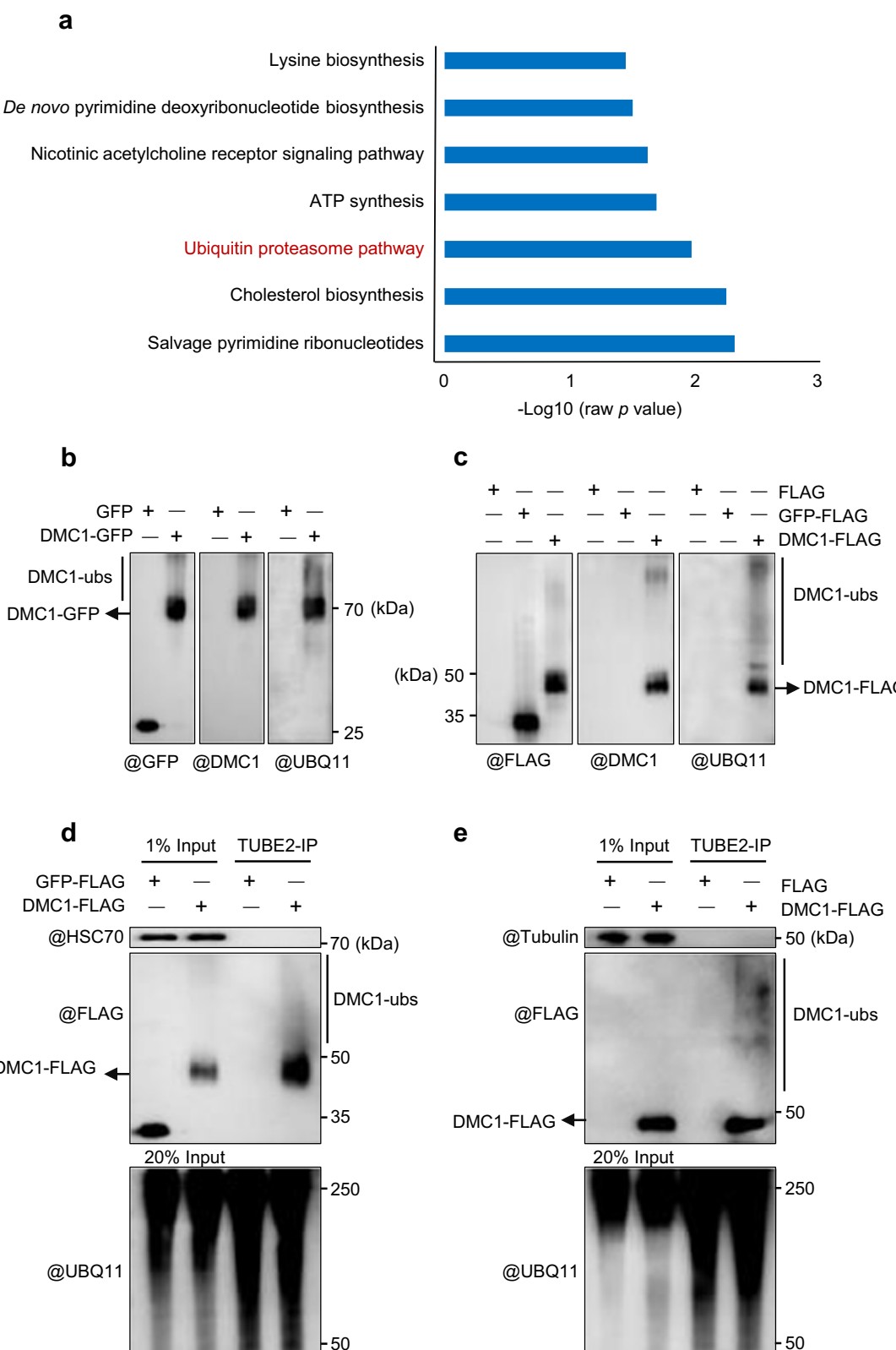

APC/C (Anaphase-Promoting Complex/Cyclosome) complex[31]; HEI10, also called CCNB1IP1 (CyCliN B1 Interacting Protein 1)[29]; and ASK1, a subunit of the SCF complex[28]. Of those, only *ask1* mutants are completely defective for meiotic recombination[28], and like *dmc1* have 10 univalent chromosomes rather than 5 paired bivalents[36]. As an SCF subunit ASK1 is expressed from leptotene to pachytene during meiosis indicating a possible function in meiotic recombination[37]. Given that

SKP1 together with CUL1, Rbx1, and a variable F-box protein forms the E3 ubiquitin ligase SCF complex[38], we focused on the approximately 700 F-box proteins in *Arabidopsis*[39]. We searched a previously published meiocyte transcriptome[40] and identified 62 F-box genes with specific or preferential expression in meiocytes (Supplementary Fig. 2a, Supplementary Data 3). We examined T-DNA alleles (see methods for details) for several of these including *At4g29420*,

**Fig. 1 | Detection of the DMC1 ubiquitination in vivo. a** Histogram shows GO term analysis of the enriched categories of biological pathways for the identified components immunoprecipitated with DMC1 following MS analysis. The −log10 (raw *p* values) of each term are taken as the abscissa. For GO term analysis, a one-tailed statistical overrepresentation test was used without adjustment for multiple comparisons. **b–e** In vivo ubiquitination assay of DMC1. **b** Anti-GFP, -DMC1, and -UBQ11 antibodies were used to examine the ubiquitination level of DMC1-GFP immunoprecipitated by anti-GFP magnetic beads from tobacco leaves infiltrated with DMC1-GFP. GFP was included as a control; **c** Anti-FLAG, -DMC1, and -UBQ11 antibodies were used to examine the ubiquitination level of DMC1-FLAG immunoprecipitated by anti-FLAG magnetic beads from central inflorescences of *Arabidopsis* *DMC1-FLAG* transgenic plants. GFP-FLAG was included as a control; **d** Anti-FLAG and -UBQ11 antibodies were used to examine the ubiquitination level of DMC1-FLAG immunoprecipitated by TUBE2 agarose beads from tobacco leaves infiltrated with DMC1-FLAG. GFP-FLAG was included as a control and HSC70 was included as an internal control; **e** Anti-FLAG and -UBQ11 antibodies were used to examine the ubiquitination level of DMC1-FLAG immunoprecipitated by TUBE2 agarose beads from central inflorescences of *Arabidopsis DMC1-FLAG* transgenic plants. FLAG was included as a control and tubulin was included as an internal control. Each experiment was repeated three times independently with similar results. Source data are provided as a Source data file.

*At2g29610*, *At3g10430*, *At2g17830*, *At5g42460*, and *At3g61730*, but none of the single mutants exhibited defects in fertility or meiosis (Supplementary Fig. 2b, c). Sequence alignment showed that AT3G61730 and AT5G36000 are homologs with 90.81% amino acid identity and have a potential N-terminal F-box domain. Interestingly, a dominant activation-tagged allele of *AT3G61730*, which is also called *Reduced Male Fertility* (*RMF*), exhibits male sterility[41], suggesting a role in meiosis. For convenience, we refer to *AT3G61730* and *AT5G36000* as *RMF1* (*Reduced Male Fertility 1*) and *RMF2* (*Reduced Male Fertility 2*), respectively. We constructed a phylogenetic tree based on multiple alinements of protein sequence using RMF1/2 and their homologs in plants (Supplementary Fig. 3). The phylogenetic tree shows that RMF1 and RMF2 are homologs of previously reported F-box proteins ZYGO1 in rice[33], and ACOZ1 in maize[34]. The tree also shows that there are three copies of *RMF* in *Arabidopsis thaliana* divided into two groups, and that *RMF1* and *RMF2* cluster with homologs in *Arabidopsis halleri* and *Arabidopsis lyrata*, while *AT3G62500*, another *RMF* gene, forms a sister lineage.

To examine the in vivo function of RMF1/2, we used a T-DNA allele (SAIL_195_A04) which we designated as *rmf1-1* (Supplementary Fig. 4a, b), and employed CRISPR-Cas9 to generate two independent alleles for *RMF2*, which we designated *rmf2-1* and *rmf2-2* (Supplementary Fig. 4c). We found that neither the *rmf1-1* nor *rmf2-1* single mutants have any developmental or fertility abnormalities (Fig. 2a, b). Interestingly, the *rmf1-1 rmf2-1* double mutant has normal vegetative growth but is completely sterile (Fig. 2a, b). Staining *rmf1-1 rmf2-1* anthers with Alexander Red showed dramatically decreased pollen viability compared to the single mutants or wild type (WT) Col-0 (Fig. 2c–f, k). Consistent with the inviable pollen phenotype, 94.22% (*n* = 89) of toluidine blue stained *rmf1-1 rmf2-1* tetrad stage meiocytes were polyads, which are indicative of abnormal meiosis, while no polyads were observed in Col-0 (Fig. 2g–j, l). We examined meiotic chromosome morphology using fluorescence in situ hybridization (FISH) with a centromere probe and found that *rmf1-1* and *rmf2-1* single mutants are indistinguishable from Col-0 during meiosis (Supplementary Fig. 4d). In contrast, the *rmf1-1 rmf2-1* double mutant has defective homologous chromosome pairing, synapsis and chiasmata formation resulting in ten univalents at diakinesis and subsequent unequal segregation of chromosomes (Fig. 2m, Supplementary Fig. 4d). To verify these results with an independent allele of *RMF1*, we used CRISPR-Cas9 to generate a 142 bp (56–197 bp) deletion in *RMF1*, which we designated *rmf1-2* (Supplementary Fig. 4a). The meiotic defects in *rmf1-2 rmf2-2* are consistent with those of *rmf1-1 rmf2-1* (Supplementary Fig. 4d). These results suggest that RMF1 and RMF2 are functionally redundant in meiosis.

## ASK1, RMF1/2, and DMC1 act in the same pathway during meiotic recombination

To test whether the recombination defect in *rmf1 rmf2* is caused by the loss of meiotic DSB formation, we used immunofluorescence to detect γH2AX foci, a DSB marker[42]. We observed no significant difference in the number of γH2AX foci at zygotene of *rmf1-1 rmf2-1* and *ask1-1* compared to Col-0 (Supplementary Fig. 5a, b), indicating that DSB formation is unaffected in *rmf1-1 rmf2-1* and *ask1-1*. We then crossed *rmf1-1 rmf2-1* and *ask1-1* with *spo11-1-1* or *mre11-4* which are defective in DSB formation and DSB repair respectively[43,44]. The *rmf1-1 rmf2-1 spo11-1-1* triple mutant and *ask1-1 spo11-1-1* double mutant have similar meiotic defects compared to *rmf1-1 rmf2-1, ask1-1*, and *spo11-1-1*, including lack of synapsis and formation of 10 univalents (Fig. 3a). The *rmf1-1 rmf2-1 mre11-4* meiocytes have chromosome entanglements and fragmentation resembling those of *mre11-4* (Fig. 3b). These findings provide additional evidence that RMF1/2 function downstream of both SPO11-1 and MRE11 during meiotic recombination.

Intriguingly, *ask1, dmc1,* and *rmf1 rmf2* have similar meiotic defects with no typical pachytene and ten univalents (Fig. 3a, c). The chromosome morphologies in *ask1, dmc1,* and *rmf1 rmf2* are indicative of failures in meiotic recombination and homolog asynapsis, so we used immunofluorescence staining to examine SYN1, the α-kleisin subunit of the *Arabidopsis* cohesin complex[45], and ZYP1A, the transverse element of the synaptonemal complex (SC)[46] in Col-0, *ask1-1*, *rmf1-1 rmf2-1*, and *dmc1-2* meiocytes. We did not observe changes in SYN1 distribution in pachytene meiocytes in any of the mutants, but ZYP1A signals are absent in *ask1-1*, *rmf1-1 rmf2-1*, and *dmc1-2* (Supplementary Fig. 5c). In addition, we generated high-order mutants and found that *ask1-1 dmc1-2*, *rmf1-1 rmf2-1 dmc1-2*, and *ask1-1 rmf1-1 rmf2-1 dmc1-2* have meiotic chromosome morphology defects similar to each of the single mutants with no additive effect (Fig. 3c). These results provided evidence that RMF1/2 and ASK1 act downstream of meiotic DSB formation as does DMC1 and that they do not have an additive effect, suggesting that they may participate in the same genetic pathway during meiotic recombination.

## RMF1/2 interact with ASK1 and DMC1 via distinct regions

RMF/RMF1 interacts with both ASK1 and ASK2 in vitro[41], and both RMF1 and RMF2 have an N-terminal F-box domain (Supplementary Fig. 6a). We used a yeast two-hybrid (Y2H) assay to show that the F-box domain of RMF2 interacts with ASK1, similar to that of RMF1 (Supplementary Fig. 6b). We confirmed the physical interaction of RMF1/2 and ASK1 using an in vitro pull-down assay (Supplementary Fig. 6c). The interaction was also validated using a split luciferase complementation (SLC) imaging assay in tobacco leaves (Supplementary Fig. 6d). We confirmed the in vivo interaction using a bimolecular fluorescent complementation (BiFC) assay (Supplementary Fig. 6e). Finally, we co-precipitated ASK1 with both RMF1/2 using an in vivo co-immunoprecipitation (co-IP) assay by expressing FLAG-tagged RMF1/2 and MYC-tagged ASK1 in tobacco leaves (Supplementary Fig. 6f). Taken together, these results demonstrate that RMF1/2 and ASK1 physically interact in vivo and may function as subunits of the same SCF complex during meiosis.

SCF complexes target substrates through their F-box protein[38], our results suggest DMC1 as a candidate target of the SCF^RMF1/2 complex. To test this hypothesis, we used an in vitro pull-down assay to show that RMF1/2 can be pulled down by DMC1 (Fig. 4a). We confirmed this interaction in vivo using a co-IP assay in tobacco leaves by expressing GFP-tagged RMF1/2 and FLAG-tagged DMC1. DMC1 was co-precipitated with both RMF1 and RMF2 (Fig. 4b). We validated the

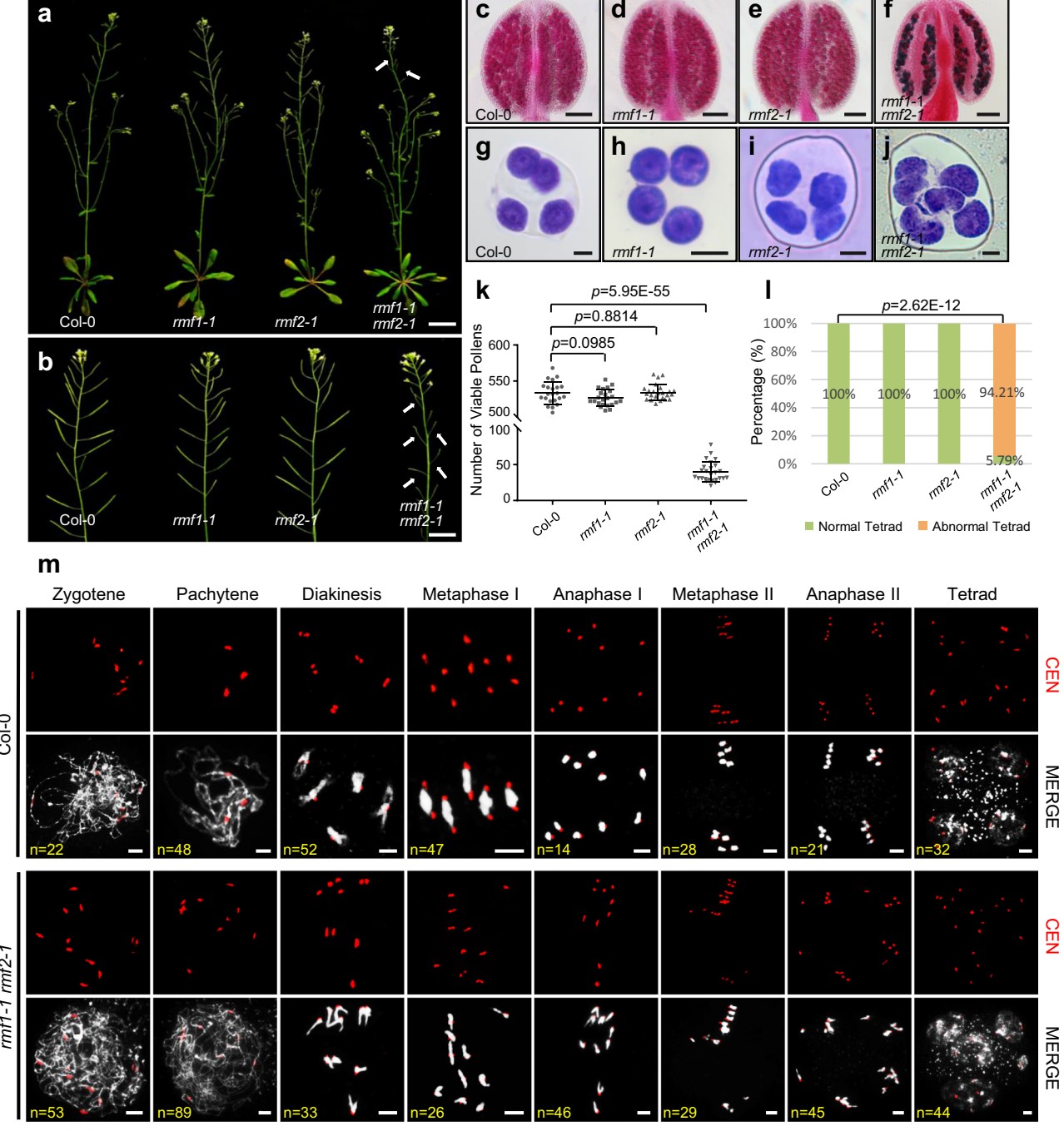

**Fig. 2 | Morphological characterization of *rmf1 rmf2*. a** Whole plant phenotypes of Col-0, *rmf1-1*, *rmf2-1*, and *rmf1-1 rmf2-1*. *rmf1-1* and *rmf2-1* plants do not show any vegetative growth or fertility abnormalities, while *rmf1-1 rmf2-1* has normal vegetative growth but is completely sterile. The white arrows point to siliques in *rmf1-1 rmf2-1*. Bar = 3 cm. **b** Siliques of Col-0 *rmf1-1*, *rmf2-1*, and *rmf1-1 rmf2-1*. The *rmf1-1* and *rmf2-1* plants have siliques similar in length to Col-0, while *rmf1-1 rmf2-1* is completely sterile. The white arrows point to siliques in *rmf1-1 rmf2-1*. Bar = 3 cm. **c**–**f** Pollen grains stained with Alexander Red from **c** Col-0, **d** *rmf1-1*, **e** *rmf2-1*, and **f** *rmf1-1 rmf2-1*. Pollen viability in *rmf1-1* and *rmf2-1* are similar to Col-0, while dramatically decreased in *rmf1-1 rmf2-1*. Bar = 100 μm. **g**–**j** Tetrad-stage microspores stained with toluidine blue from **g** Col-0, **h** *rmf1-1*, **i** *rmf2-1*, and **j** *rmf1-1 rmf2-1*. *rmf1-1* and *rmf2-1* has normal tetrads similar to Col-0, while *rmf1-1 rmf2-1* has polyads. Bar = 5 μm. **k** Scatter plot of viable pollen number. Pollen viability in *rmf1-1* and *rmf2-1*

have no significant difference compared with Col-0, while significantly decreased in *rmf1-1 rmf2-1*. Data are presented as the mean values ± SD, *p* values were calculated using a two-tailed Student's *t*-test. For pollen grain viability analysis, 22 anthers for Col-0, 23 anthers for *rmf1-1*, 24 anthers for *rmf2-1*, 26 anthers for *rmf1-1 rmf2-1* isolated from more than three independent plants were used. **l** Histogram of the proportion of normal versus abnormal tetrads. Abnormal tetrads were observed in *rmf1-1 rmf2-1*. Data are presented as the mean values ± SD, *p* values were calculated using a two-tailed Student's *t*-test. **m** Meiotic chromosome phenotypes of Col-0 and *rmf1-1 rmf2-1* assayed by centromere FISH. *rmf1-1 rmf2-1* is defective in typical pachytene chromosome and bivalent formation. Bar = 5 μm. For each meiotic stage in Col-0 and *rmf1-1 rmf2-1*, cells isolated from more than three independent plants were observed with similar meiotic chromosome phenotypes. The number of cells observed was labeled in the figures. Source data are provided as a Source data file.

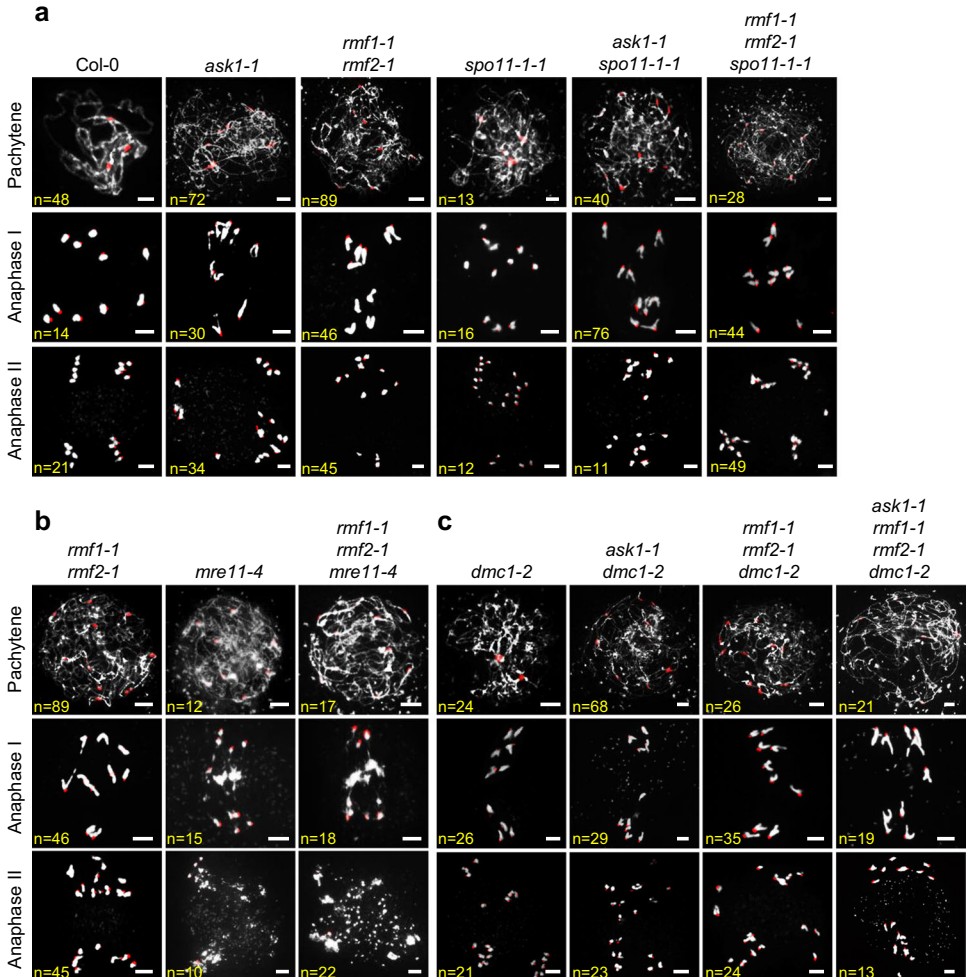

**Fig. 3 | Genetic analyses of *ASK1* and *RMF1/2* with genes related to meiotic recombination.** Meiotic chromosome morphology at pachytene, anaphase I, anaphase II chromosomes assayed by centromere FISH. Bar = 5 μm. **a** The *spo11-1-1* single mutant has no typical pachytene-like chromosome morphology and instead has ten univalents, and *ask1-1 spo11-1-1* and *rmf1-1 rmf2-1 spo11-1-1* have meiotic chromosome morphology similar to *spo11-1-1*. **b** The *rmf1-1 rmf2-1* double mutant has no typical pachytene-like chromosome morphology and instead has ten univalents without chromosomes entanglements and fragments, while *rmf1-1 rmf2-1*

*mre11-4* triple mutant has severe chromosomes entanglements and fragmentation similar to *mre11-4*. **c** The *ask1-1 dmc1-2*, *rmf1-1 rmf2-1 dmc1-2*, and *ask1-1 rmf1-1 rmf2-1 dmc1-2* higher-order mutants have meiotic recombination defects similar to *dmc1-2*, *ask1-1*, and *rmf1-1 rmf2-1*. For each meiotic stage in above-mentioned plants, cells isolated from more than three independent plants were observed with similar meiotic chromosome phenotypes. The number of cells observed was labeled in the figures.

in vivo interaction using a BiFC assay and observed strong nuclear signals in tobacco cells (Fig. 4c). We also used the SLC imaging assay in tobacco and observed that both RMF1 and RMF2 interact with DMC1 but not RAD51 (Fig. 4d), suggesting that RMF1/2 specifically binds to DMC1 rather than RAD51. To determine which regions of RMF1/2 interact with DMC1, we expressed truncated forms of RMF1/2 and DMC1 using the SLC imaging assay. We observed the strongest LUC signal from interactions between DMC1 and RMF1/2-C, rather than RMF1/2-N (Supplementary Fig. 7a, b). Furthermore, DMC1-C interactions with RMF1/2-C, rather than DMC1-N, produce the strongest LUC activities (Supplementary Fig. 7a, c, d), suggesting that RMF1/2 interact with the DMC1-C-terminus through their C-termini. Taken together, our results demonstrate that RMF1/2 interact with ASK1 and DMC1 using distinct regions and these interactions probably link the SCF^RMF1/2 complex to its target DMC1 during meiosis.

## DMC1 can substitute for RAD51 in the absence of ASK1 and/or RMF1/2

Given that RMF1/2 specifically interact with DMC1 but not RAD51, we examined the relationship of RMF1/2 and ASK1 with RAD51. Unlike

*dmc1*, the *rad51-1* mutant has severe chromosome fragmentation and entanglements during meiosis[47]. Unexpectedly, *rmf1-1 rmf2-1 rad51-1*, *ask1-1 rad51-1*, and *ask1-1 rmf1-1 rmf2-1 rad51-1* have much milder chromosome entanglement and fragmentation (Fig. 5a). Examination of high-order mutants that include *dmc1-2* found that the milder phenotype is dependent on DMC1 (Fig. 5b), suggesting that DMC1 can substitute for RAD51 in the absence of either ASK1 and/or RMF1/2. To test this hypothesis, we immunostained meiocytes to examine the DMC1 foci in those mutants. We observed significantly more DMC1 foci in zygotene meiocytes of *rmf1-1 rmf2-1* and *ask1-1* compared with Col-0 (Fig. 5c, Supplementary Fig. 8a). In Col-0 meiocytes DMC1 foci experience a reduction in numbers from zygotene to pachytene, but in *rmf1-1 rmf2-1* and *ask1-1* the elevated DMC1 foci numbers persisted in pachytene-like meiocytes (Fig. 5c, Supplementary Fig. 8a), indicating that ASK1 and RMF1/2 are required to remove DMC1 from chromosomes. Interestingly, we also found that both *rmf1-1 rmf2-1 rad51-1* and *ask1-1 rad51-1* have significantly more DMC1 foci compared to *rad51-1* (Fig. 5c, Supplementary Fig. 8a), which is consistent with the alleviation of the meiotic chromosome fragmentation and entanglement phenotypes

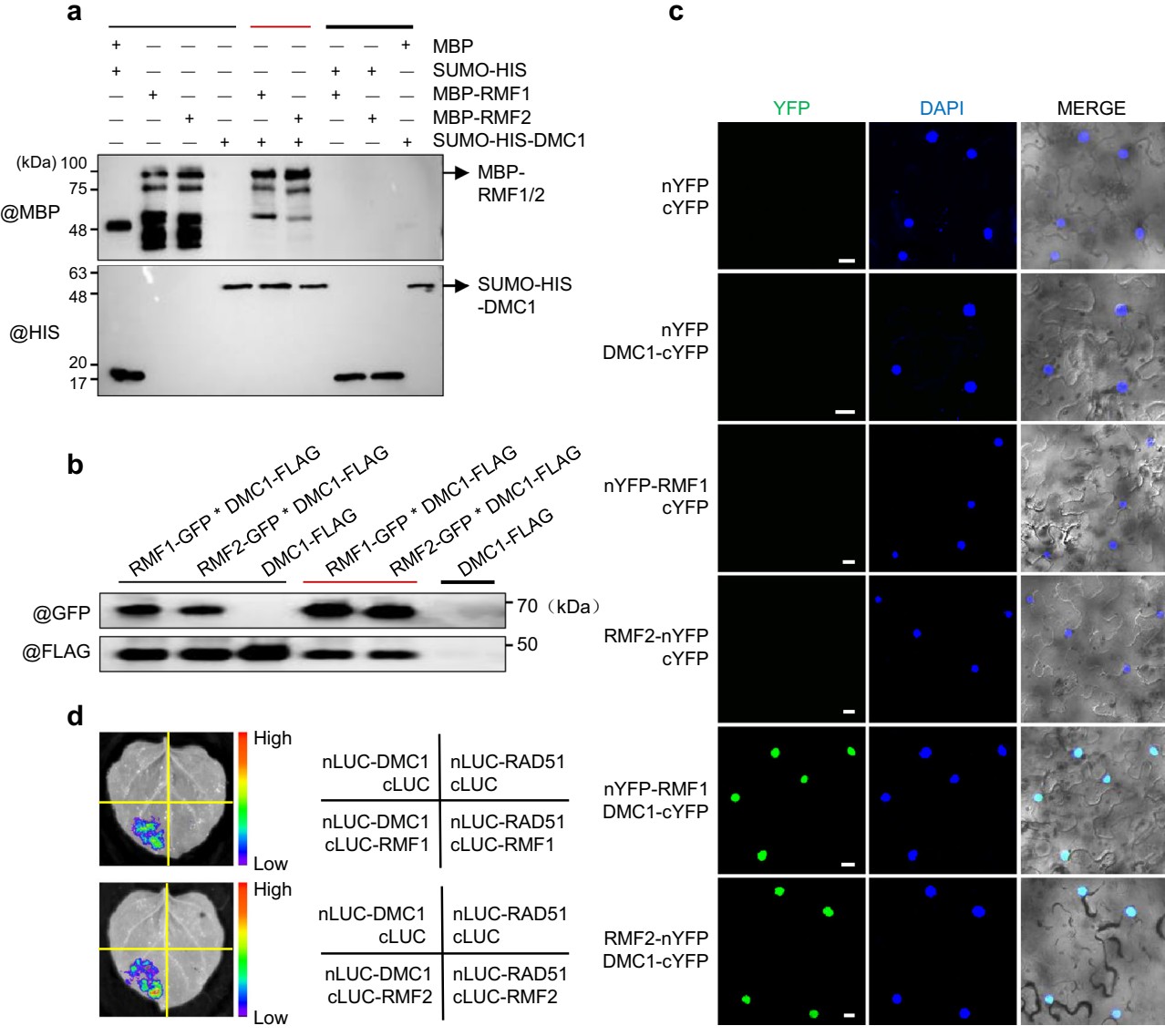

**Fig. 4 | RMF1/2 interact with DMC1 both in vitro and in vivo. a** Interaction assay between RMF1/2 and DMC1 by pull-down. The recombinant MBP-RMF1/2 and SUMO-HIS-DMC1 were heterologously expressed in *E. coli*. SUMO-HIS-DMC1 with MBP-RMF1 or MBP-RMF2 were pulled-down using HIS Resin and examined with anti-MBP and -HIS antibodies. Black lines indicate input (thin line) and negative control (thick line), and red line indicates the experimental group. This experiment was repeated three times independently with similar results. **b** Validation of the RMF1/2-DMC1 interaction by transient expression and co-IP in tobacco leaves. Anti-FLAG and -GFP antibodies were used to detect the precipitates immunoprecipitated by anti-GFP magnetic beads from tobacco leaves co-infiltrated with RMF1-GFP/DMC1-FLAG and RMF2-GFP/DMC1-FLAG. DMC1-FLAG was included as a control. Black lines indicate input (thin line) and negative control (thick line), and red line indicates the experimental group. This experiment was repeated three times independently with similar results. **c** Validation of the RMF1/2-DMC1 interaction by bimolecular fluorescence complementation (BiFC) assay in tobacco leaves. RMF1/2

were fused to an N-terminal fragment of YFP (nYFP) and DMC1 was fused to a C-terminal fragment of YFP (cYFP). Paired proteins were co-infiltrated into tobacco leaves with nYFP or cYFP as negative control. Nuclear signals were observed in tobacco cells co-infiltrated with nYFP-RMF1/DMC1-cYFP and RMF2-nYFP/DMC1-cYFP. Bar = 20 μm. This experiment was repeated three times independently with similar results and 30 tobacco nuclei were observed with nuclear signals for each combination. **d** Split luciferase complementation imaging assays was used to examine the interactions between RMF1/2 with DMC1 and RAD51. DMC1 and RAD51 were fused to an N-terminal fragment of luciferase (nLUC), RMF1 and RMF2 were fused to a C-terminal fragment of luciferase (cLUC). Paired proteins were infiltrated into tobacco leaves with nLUC or cLUC as negative control. The right panel displays each combination. Robust LUC activities were observed in areas co-infiltrated with nLUC-DMC1/cLUC-RMF1 and nLUC-DMC1/cLUC-RMF2. Source data are provided as a Source data file.

in the same backgrounds (Fig. 5a). We also analyzed the relationship of RMF1/2 and two RAD51 auxiliary factors RAD51C and XRCC3, that have chromosome entanglement and fragmentation phenotypes similar to *rad51* when mutated[48]. We found that *rmf1-1 rmf2-1 rad51c* and *rmf1-1 rmf2-1 xrcc3* triple mutants have reduced chromosome fragmentation, similar to *rmf1-1 rmf2-1 rad51-1*, compared with *rad51c* and *xrcc3* single mutants (Supplementary Fig. 8b). In addition, SYN1 signals are indistinguishable in pachytene meiocytes from

*rmf1-1 rmf2-1 rad51-1*, *ask1-1 rad51-1*, and *ask1-1 rmf1-1 rmf2-1 rad51-1*, but ZYP1A signals are absent (Supplementary Fig. 5c), suggesting a synapsis defect. Given the observation of univalents in these mutants, we speculate that DMC1 may be able to substitute for RAD51 function in inter-sister recombination. These results indicate that DMC1 can substitute for RAD51 during meiotic DSB repair, perhaps using sister chromatids as templates, in the absence of ASK1 and/or RMF1/2.

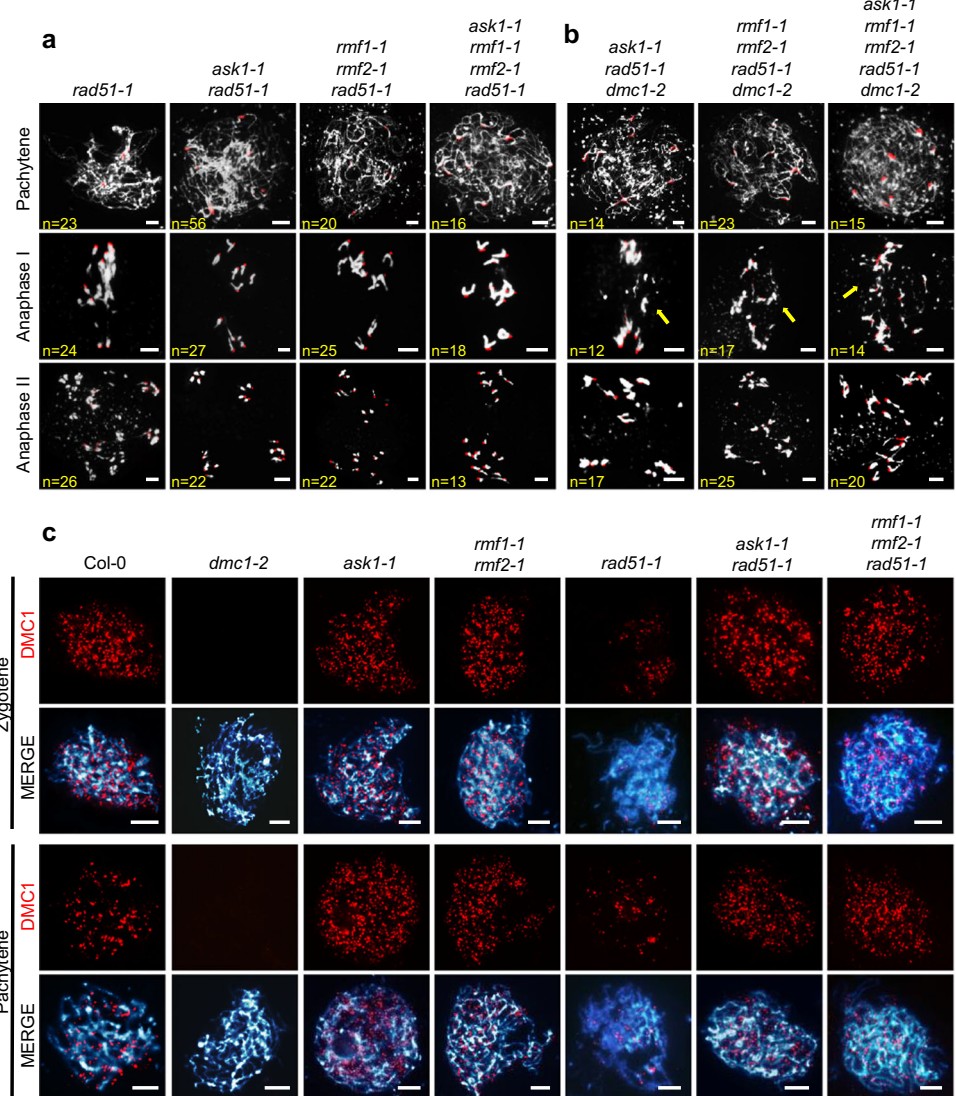

**Fig. 5 | DMC1 can substitute for RAD51 in the absence of ASK1 and/or RMF1/2.**
**a** The *rad51-1* single mutant has severe chromosomes entanglements and fragmentation, which are dramatically decreased in the *ask1-1 rad51-1*, *rmf1-1 rmf2-1 rad51-1*, and *ask1-1 rmf1-1 rmf2-1 rad51-1* higher-order mutants. **b** The chromosome entanglement and fragmentation phenotypes in the higher-order mutants are dependent on DMC1. The yellow arrows in each figure point out the entanglements and fragments. For each meiotic stage in above-mentioned plants, cells isolated from more than three independent plants were observed with similar meiotic chromosome phenotypes. The number of cells observed was labeled in the figures. **c** Localization of DMC1 foci in Col-0, *dmc1-2, ask1-1, rmf1-1 rmf2-1, rad51-1, ask1-1 rad51-1, rmf1-1 rmf2-1 rad51-1* at zygotene and pachytene meiocytes. More DMC1

foci at zygotene and persistent DMC1 foci at pachytene-like stage are observed in *rmf1-1 rmf2-1* and *ask1-1* compared with Col-0. More DMC1 foci at zygotene and pachytene-like stage are observed in *ask1-1 rad51-1* and *rmf1-1 rmf2-1 rad51-1* compared with *rad51-1*. Bar = 5 μm. For DMC1 foci number analysis, 22 cells for Col-0 zygotene, 22 cells for Col-0 pachytene, 23 cells for *dmc1-2* zygotene, 20 cells for *dmc1-2* pachytene, 20 cells for *ask1-1* zygotene, 20 cells for *ask1-1* pachytene, 22 cells for *rmf1-1 rmf2-1* zygotene, 21 cells for *rmf1-1 rmf2-1* pachytene, 20 cells for *rad51-1* zygotene, 25 cells for *rad51-1* pachytene, 20 cells for *ask1-1 rad51-1* zygotene, 21 cells for *ask1-1 rad51-1* pachytene, 21 cells for *rmf1-1 rmf2-1 rad51-1* zygotene, 21 cells for *rmf1-1 rmf2-1 rad51-1* pachytene isolated from more than three independent plants were observed.

## RMF1/2 ubiquitinate DMC1 for subsequent degradation by the 26S proteasome

To examine whether RMF1/2 are required for the ubiquitination of DMC1, we conducted an in vitro ubiquitination assay using recombinant DMC1 heterologously expressed and purified from *E. coli* as substrate. We immunoprecipitated SCF[RMF1/2] complexes using anti-FLAG antibody against extracts from RMF1/2-FLAG transgenic plants, and incubated captured SCF[RMF1/2] complexes with E1 (UBE1 from yeast), E2 (UbcH5c from human), recombinant HIS-ubiquitin, and SUMO-HIS-DMC1. DMC1 is ubiquitinated in the presence of E1-E2 with or without SCF[RMF1/2] as E3 (Supplementary Fig. 9a). Considering that E2 alone is sufficient to transfer ubiquitin to substrates[49], our result demonstrates that DMC1 can be ubiquitinated in vitro. To investigate the effect of

RMF1/2 on DMC1's ubiquitination in vivo, we examined the ubiquitination level of DMC1 by immunoprecipitating DMC1 from *DMC1-FLAG/Col-0* and *DMC1-FLAG/rmf1-1 rmf2-1* transgenic plants using TUBE2 agarose beads. DMC1 is significantly less ubiquitinated in *rmf1-1 rmf2-1* compared with Col-0 (Fig. 6a), supporting a role for RMF1/2 in promoting DMC1's ubiquitination in vivo.

Protein ubiquitination, especially poly-ubiquitination, usually enables the substrates to be degraded by the 26S proteasome[21]. To test whether DMC1 ubiquitination leads to its proteasomal degradation, we examined DMC1 levels in plant tissue treated with the 26S proteasome inhibitor MG-132 versus control tissue. Using a transient DMC1 tobacco leaf expression system, leaves treated with MG-132 showed significantly higher levels of DMC1 compared with those treated with

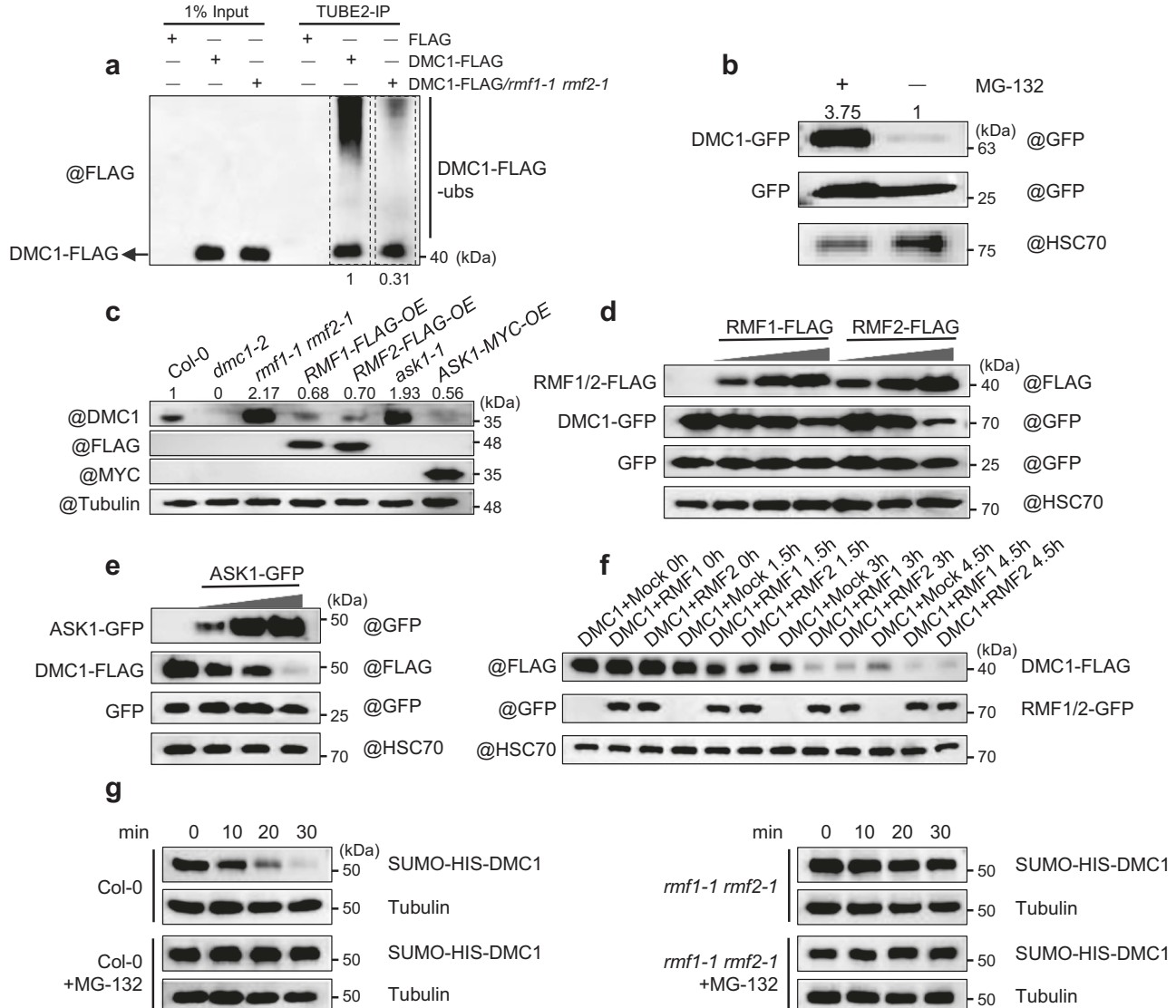

**Fig. 6 | RMF1/2 facilitate the DMC1's ubiquitination and proteasomal degradation. a** RMF1/2 are responsible for DMC1 ubiquitination in vivo. Anti-FLAG antibody was used to examine the ubiquitination level of DMC1-FLAG immunoprecipitated by TUBE2 agarose beads from central inflorescences of *DMC1-FLAG* and *DMC1-FLAG/rmf1-1 rmf2-1* transgenic *Arabidopsis* (validated in Supplementary Fig. 1a–c). FLAG was included as a control. **b** Treatment with MG-132 increases DMC1 stability. Anti-GFP antibody was used to examine the protein level of DMC1-GFP in tobacco leaves co-infiltrated with DMC1-GFP and GFP treated with or without MG-132. **c** The protein level of DMC1 is higher in *rmf1-1 rmf2-1* and *ask1-1* mutants and lower in *RMF1-*, *RMF2-*, and *ASK1-* overexpressing (*OE*) plants compared with Col-0. Anti-DMC1 antibody was used to examine the protein level of DMC1 in central inflorescences, with *dmc1-2* as a negative control. **d**, **e** (**d**) RMF1/2 or (**e**) ASK1 promotes DMC1 degradation in vivo. Anti-FLAG and -GFP antibodies were used to examine the protein levels of **d** RMF1/2-FLAG and DMC1-GFP or **e** ASK1-GFP and DMC1-FLAG in tobacco leaves co-infiltrated with **d** GFP, DMC1-GFP and different

amount of RMF1/2-FLAG or **e** GFP, DMC1-FLAG and different amount of ASK1-GFP. GFP was included as a control in (**b**, **d**, **e**). **f** RMF1/2 promote DMC1 degradation in a semi-in vivo system. Anti-FLAG and -GFP antibodies were used to examine the protein levels of DMC1-FLAG and RMF1/2-GFP in samples mixed with RMF1/2-GFP or un-infiltrated WT tobacco leaves with DMC1-FLAG and incubated at room temperature for the corresponding time. **g** Degradation rate of recombinant SUMO-HIS-DMC1 in Col-0 and *rmf1-1 rmf2-1* measured using cell-free degradation assay. Anti-HIS antibody was used to examine the protein level of SUMO-HIS-DMC1 in samples of SUMO-HIS-DMC1 mixed with equal amount of Col-0 or *rmf1-1 rmf2-1* crude protein collected from central inflorescences and incubated at room temperature for the corresponding time, treated with or without MG-132. HSC70 (**b**, **d**–**f**) or Tubulin (**c**, **g**) were included as an internal control. The numbers above or below corresponding band indicate the relative protein levels (**a**–**c**). Each experiment was repeated three times independently with similar results. Source data are provided as a Source data file.

MgCl₂ as a control (Fig. 6b), indicating that inhibition of 26S proteasome-dependent degradation has a notable effect on DMC1 stability. To confirm that DMC1 stability is regulated by SCF$^{RMF1/2}$ in vivo, we examined DMC1 levels in different mutant backgrounds using western blots probed with an anti-DMC1 antibody, and observed significantly higher DMC1 levels in *rmf1-1 rmf2-1*, and *ask1-1* mutants compared with Col-0, but dramatically less in *RMF1/2-OE* and *ASK1-OE* plants relative to Col-0 (Fig. 6c, Supplementary Fig. 9b–j), further

demonstrating that RMF1/2 and ASK1 are responsible for the degradation of DMC1.

We then evaluated the dynamics of DMC1's degradation mediated by RMF1/2. With increasing amounts of RMF1/2, DMC1 shows a gradual decrease in vivo (Fig. 6d). In addition, ASK1 can also promote the degradation of DMC1 in vivo (Fig. 6e). We next used a semi-in vivo E3 ligase-promoted substrate degradation assay to detect the RMF1/2-mediated DMC1 degradation in a time course experiment. DMC1

exhibits gradual reduction in response to either RMF1 or RMF2 (Fig. 6f). We validated these results with an independent cell-free degradation assay system using recombinant DMC1 mixed with proteins extracted from central inflorescences from Col-0 or *rmf1 rmf2* with or without MG-132 treatment (Fig. 6g). Notably, both MG-132 and mutation of *RMF1/2* suppress DMC1's degradation (Fig. 6g). The tagged versions of proteins may not be able to complement full biological function of their wild-type counterparts, but taken as a whole, the experiments using tagged proteins are consistent with the finding that ASK1-RMF1/2 facilitate DMC1's degradation. Together, these results provide strong evidence that SCF^RMF1/2 facilitates DMC1 degradation via the ubiquitin-proteasome system.

### Identification and functional characterization of DMC1 ubiquitination sites

To characterize the mechanism of DMC1 ubiquitination, we used a mass spectrometry assay to identify five DMC1 lysine residues (K45, K70, K101, K162, and K290) as potential ubiquitination sites (Supplementary Table 1). In addition, we found that K46 adjacent to K45 is highly conserved in eukaryotes (Supplementary Fig. 10a). Among them, K45, K46, and K70 are located in the HhH (helix-hairpin-helix) nonspecific DNA-binding motif, while K101, K162, and K290 are embedded in the RecA/RAD51 domain (Supplementary Fig. 10a). In addition, K46, K70, K162, and K290 are highly conserved in plants (*Arabidopsis*, rice and maize), animals (human, mouse, and *C. elegans*), and fungi (budding yeast and fission yeast), while K45 and K101 are less conserved in these species (Supplementary Fig. 10a). To test the function of the six sites, we mutated all six lysines (K) into arginines (R) (designated DMC1-6KR hereafter). Using the AlphaFold protein structure database[50], we predicted DMC1 and DMC1-6KR protein structures, and found that both DMC1 and DMC1-6KR show similar protein structure (Supplementary Fig. 10b, c). Consistent with these observations, DMC1 and DMC1-6KR have the same nuclear localization in tobacco cells (Fig. 7a), and interaction with RMF1/2 (Fig. 7b).

We employed an in vivo ubiquitination assay using transient expression of DMC1 and DMC1-6KR fused with FLAG or GFP to examine whether DMC1-6KR is competent for ubiquitination. Compared with wild-type DMC1, samples infiltrated with DMC1-6KR showed significant attenuation of the ubiquitinated bands (Fig. 7c, d), indicating that the six lysine residues are required for DMC1 ubiquitination. We used an in vivo degradation assay in tobacco leaves to test whether mutating the lysine residues changed the degradation of DMC1-6KR compared to DMC1. We observed that the degradation of DMC1-6KR tissues treated with or without MG-132 did not significantly change, while MG-132 treatment protected wild-type DMC1 from degradation (Fig. 7e). A similar result was observed using an in vitro cell-free degradation assay. Recombinant DMC1-6KR showed an attenuated degradation rate compared with DMC1 (Fig. 7f). These results support a role for the six lysine resides in mediating the ubiquitination of DMC1 and its subsequent proteasomal degradation.

To further validate the critical role of DMC1 ubiquitination in meiosis in vivo, we performed trans-complementation with intact DMC1 and DMC1-6KR. A recent study showed that the untagged DMC1 driven by the *RAD51* promoter (*pRAD51::DMC1g*) can restore the meiotic defects in *dmc1* mutants[51]. We then generated full-length *pRAD51-DMC1g/dmc1* and *pRAD51-DMC1g-6KR/dmc1* transgenic plants in the *dmc1-2* mutant background (Supplementary Fig. 11a). Consistent with the previous study, the fertility and meiotic defects were restored in *pRAD51::DMC1g/dmc1* transgenic lines, but not in the *pRAD51::DMC1g-6KR/dmc1* lines (Supplementary Fig. 11b). We then selected two independent lines of *pRAD51::DMC1g/dmc1* and two *pRAD51::DMC1g-6KR/dmc1* plants for quantifying the expression level of *DMC1*. We observed no significant difference in *DMC1* expression between *pRAD51::DMC1g/dmc1* and *pRAD51::DMC1g-6KR/dmc1* transgenic lines (Supplementary Fig. 11c, d). However, the DMC1 protein

level was slightly increased in *pRAD51::DMC1g/dmc1* compared with Col-0, and significantly increased in *pRAD51::DMC1g-6KR/dmc1* compared with *pRAD51::DMC1g/dmc1* (Supplementary Fig. 11e). These results indicate that the K to R mutation in DMC1 does not complement the *dmc1*'s meiotic defects. Together, we propose that loss of ubiquitination of DMC1 might leads to meiotic recombination defects and the proper level of DMC1 is critical for its function in meiosis.

## Discussion

The SCF (SKP1-CULLIN1-F-box) complex is the largest E3 ubiquitin ligase and is pivotal for ubiquitination-mediate protein degradation[38]. Within the SCF complex, the F-box subunit determines the substrate specificity for subsequent ubiquitin-mediated degradation[38]. *Arabidopsis* has nearly 700 F-box proteins that play diverse roles in growth, development, and stress responses[39,49]. Only three F-box proteins, MOF and ZYGO1 in rice[32,33], and ACOZ1 in maize[34], are known to be required for meiosis in plants. Mutation of either *ZYGO1* or *ACOZ1* causes meiotic recombination defects and a failure to form bivalents[33,34]. Although they all interact with the ASK1/SKP1 orthologs, the function of these meiotic SCF complexes has not been investigated. We provide evidence that ASK1-RMF1/2 are components of a meiotic SCF E3 complex in *Arabidopsis*. First, like other F-box proteins, RMF1/2 interact with ASK1 through its F-box domain. Second, *rmf1 rmf2* double mutant exhibits meiotic recombination defect similar to *ask1*. Third, genetic analysis shows that RMF1/2 and ASK1 function in the same meiotic recombination pathway. Given that RMF1/2, ZYGO1 in rice and ACOZ1 in maize are close homologs, with similar meiotic defects in their corresponding mutants, it is likely that OsZYGO1 and ZmACOZ1 regulate meiotic recombination via a similar mechanism to RMF1/2. Although RMF1 interacts with both ASK1 and ASK2 in vitro[41], *ask1* and *ask2* have functional redundancy in multiple developmental processes, but only ASK1 is required for meiosis[52]. Therefore, we speculate that RMF1/2 targeting DMC1 might require specific association with ASK1 rather than ASK2.

DMC1 in mice and humans is ubiquitinated in preparation for degradation[18,20]. Consistent with these discoveries, our results suggest that *Arabidopsis* DMC1 is ubiquitinated both in vivo and in vitro, leading to its proteasomal degradation. Thus, we suggest that ubiquitination-mediated proteasomal degradation of DMC1 is conserved among eukaryotes. Since different E3 ligases are responsible for RAD51 ubiquitination[16,18,19], it is possible that DMC1 may also be targeted by other E3 ligases as well. It is also possible that employing different E3 ligases to degrade DMC1 and RAD51 might be caused by different spatiotemporal expression patterns and might assist in differentiating their functions as recombinases.

Previous studies support the idea that DMC1 is important for enforcing inter-homolog (IH) bias during meiotic recombination, while RAD51 acts as an accessory factor for DMC1-mediated strand exchange[53]. In addition, evidences suggest that DMC1 is indispensable for repairing meiotic DSBs, while the catalytic activity of RAD51 is essential for repairing somatic DSBs but not meiotic ones. For example, the catalytically inactive *rad51-II3A* yeast mutant and RAD51-GFP protein in *Arabidopsis* are incapable of repairing somatic DNA breaks, but are sufficient for meiotic DSB repair[54,55]. However, *dmc1-II3A* cannot form joint-molecules due to lack recombination activity, and has a meiotic arrest phenotype similar to the *dmc1Δ* mutant[54], indicating that DMC1 is sufficient for repairing meiotic DSBs in RAD51 catalytically defective mutants. A recent study showed that SMC5/6 can suppress *rad51* meiotic defects in a DMC1-dependent manner[56], further supporting the idea that DMC1 is capable of repairing all DSBs in the absence of RAD51. In *Arabidopsis*, introducing *dmc1-2* into the *rad51-2* knock-down allele alleviates non-homologous associations and aggravates chromosomes fragmentation[36], suggesting that DMC1 is responsible for repairing some DSBs in *rad51-2*, and RAD51 is required for DMC1 IH bias. A similar phenomenon was also observed in yeast[57].

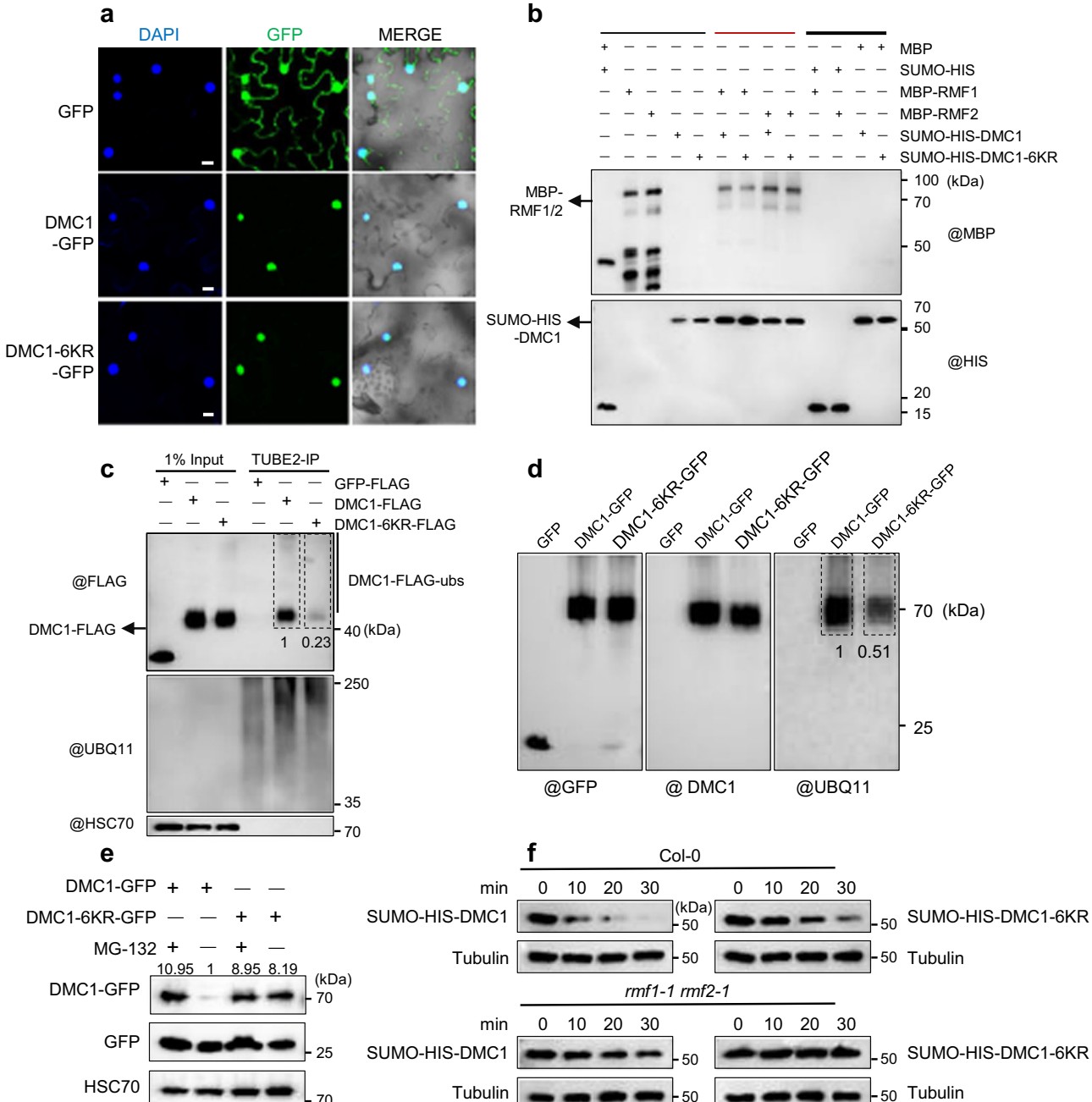

**Fig. 7 | Identification and functional characterization of DMC1 ubiquitination sites. a** DMC1 and DMC1-6KR have similar subcellular localization. DMC1-GFP and DMC1-6KR-GFP were infiltrated into tobacco leaves and strong signals were observed in nuclei. GFP was included as a control. Bar = 20 μm. This experiment was repeated three times independently with similar results and 35 tobacco nuclei were observed with nuclear signals for DMC1 and DMC1-6KR. **b** Validation of the interaction between DMC1-6KR and RMF1/2 by pull-down assay. The recombinant MBP-RMF1/2 and SUMO-HIS-DMC1/DMC1-6KR were heterologously expressed in *E. coli*. SUMO-HIS-DMC1/DMC1-6KR with MBP-RMF1/2 were pulled-down using HIS Resin and examined with anti-MBP and -HIS antibodies. Black lines indicate input (thin line) and negative control (thick line), and red line indicates the experimental group. **c**, **d** DMC1-6KR has decreased ubiquitination levels in tobacco compared with DMC1. Anti-GFP, -FLAG, -UBQ11, and -DMC1 antibodies were used to examine the ubiquitination levels of DMC1 and DMC1-6KR from tobacco leaves immunoprecipitated by **c** TUBE2 agarose beads and **d** anti-GFP magnetic beads,

respectively. HSC70 was included as an internal control. **e** The inhibition of DMC1-6KR degradation by MG-132 is alleviated relative to DMC1. Anti-GFP antibody was used to examine the protein levels of DMC1-GFP and DMC1-6KR-GFP in tobacco leaves co-infiltrated with DMC1 or DMC1-6KR and GFP, and treated with or without MG-132. GFP was included as a control and HSC70 was included as an internal control. **f** Degradation rate of recombinant SUMO-HIS-DMC1-6KR was alleviated compared with SUMO-HIS-DMC1 in cell-free degradation assay. Anti-HIS antibody was used to examine the protein levels of SUMO-HIS-DMC1 and SUMO-HIS-DMC1-6KR in samples of recombinant SUMO-HIS-DMC1 or SUMO-HIS-DMC1-6KR mixed with equal amounts of Col-0 or *rmf1-1 rmf2-1* crude proteins collected from *Arabidopsis* central inflorescences and incubated at room temperature for the corresponding time. Tubulin was included as an internal control. Each experiment was repeated three times independently with similar results. Source data are provided as a Source data file.

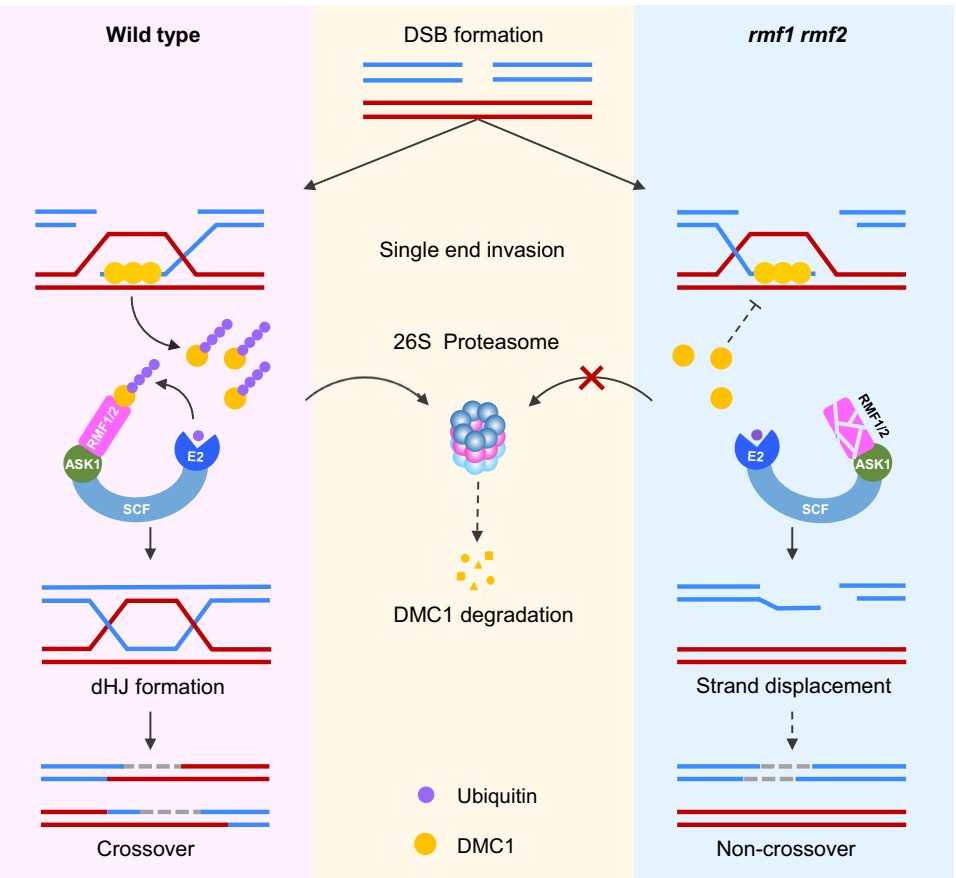

**Fig. 8 | A Model for ubiquitination of DMC1 by the SCF^RMF complex during meiotic recombination.** Meiotic recombination is initiated by the formation of DSB, which are processed to yield 3′ overhangs of single-stranded DNA (ssDNA). DMC1 facilitates the invasion of the ssDNA into homologous non-sister chromatids to form a D-loop. In wild type, the removal of DMC1 is mediated by RMF1/2 via interaction by its non-F-box domain, and SCF^RMF1/2 facilitates the ubiquitination of DMC1 for degradation by the 26S proteasome. In the absence of RMF1/2 or ASK1, DMC1 fails to be ubiquitinated and degraded and its removal from the ssDNA is compromised. The over-accumulation of DMC1 may create negative feedback to inhibit its disassembly from ssDNA, which impedes the action of DNA polymerase and prevents subsequent steps in the recombination pathway, thereby causing failure of recombination.

In addition, RAD51's role in recruiting DMC1 onto ssDNA is conserved among animals, fungi and plants[58–60]. Taken together, DMC1 alone is sufficient for both inter-sister and inter-homolog recombination during meiosis, while the role of RAD51 in meiotic recombination is independent of its catalytic activity.

In this study, our higher-order mutants analysis shows that the absence of RMF1/2 and/or ASK1 dramatically decreases meiotic chromosome fragmentation in *rad51*. In addition, the loss of DMC1 foci observed in a *rad51* background is reversed in higher-order mutant combinations with *rmf1 rmf2* or *ask1*. Thus, we hypothesize that, in the *rad51* null-mutant, DMC1 is significantly reduced due to the activity of SCF^RMF1/2, and the residual DMC1 has compromised IH bias which leads to chromosome fragmentation and entanglement. In the absence of RMF1/2 and/or ASK1 in a *rad51* background, DMC1 degradation is compromised enabling it to substitute for RAD51 and repair DSBs using sister chromatids as templates, only inter-sister DSBs were repaired in *rmf1 rmf2 rad51* and *ask1 rad51* mutants. In contrast, when RAD51 is present, *rmf1 rmf2* or *ask1* mutants allow DMC1 to inappropriately accumulate and compromise IH bias-mediated meiotic recombination.

Based on previous findings and this study, we proposed a model in which RMF1/2 regulate meiotic recombination by targeting DMC1 for ubiquitination via proteasomal degradation (Fig. 8). In the model, following DSB formation and end resection, DMC1 binds to 3′ ssDNA to facilitate strand invasion of homologous non-sister chromatids. DMC1 is then removed to allow DNA polymerase to catalyze recombination-associated DNA synthesis. We provide evidence that, in wild type, DMC1 is targeted by SCF^RMF1/2, ubiquitinated, and then degraded by the 26S proteasome. In the absence of RMF1/2 or ASK1, the removal of DMC1 from the ssDNA is attenuated due to the failure of ubiquitination and protein degradation. The over-accumulation of DMC1 may create negative feedback to inhibit its disassembly from ssDNA, which impedes the replacement of DMC1 by DNA polymerase, which in turn prevents subsequent steps in the recombination pathway such as D-loop extension or second-end capture, thereby causing failure of the double Holliday junction (dHJ) formation.

In summary, the precise assembly and proper degradation of DMC1 are stringently regulated, and the mechanism may be highly conserved from yeast to humans. Therefore, we speculate that the molecular mechanisms that regulate ubiquitination of DMC1 identified here might be a pan-eukaryotic mechanism during meiotic recombination. These discoveries provide unexplored insight to our understanding of meiotic recombination. However, the underlying mechanism how RMF1/2-mediated degradation of DMC1 in meiocytes to maintain a proper level, thus ensuring meiotic inter-homolog recombination needs further investigation.

## Methods
### Plant materials and growth conditions
*Arabidopsis* Wild-type (WT), mutants, and transgenic plants used in this study were all in the Col-0 ecotype background, with the exception of *spo11-1-1* in the Wassilewskija (Ws) ecotype, and *ask1-1* in the

Landsberg *erecta* (L*er*) ecotype. The *rmf1-2, rmf2-1,* and *rmf2-2* alleles were generated by CRISPR-cas9 following previously published procedures[61]. At4g29420 (SALK_002952), At2g29610 (SAIL_171_C06), At3g10430 (SALK_011749), At2g17830 (GABI_425C11), At5g42460 (GABI_204B06), rmf1-1 (SAIL_195_A04), ask1-1[28], spo11-1-1[43], dmc1-2 (SAIL_170_F08)[36], rad51-1 (GABI_134A01)[47], rad51c (SALK_021960)[48], xrcc3 (SALK_045564)[48], mre11-4 (SALK_028450)[44] used in this study were genotyped with PCR primers as described in Supplementary Data 4. The higher-order mutants were generated by crossing corresponding mutants mentioned above. *DMC1-FLAG* under control of the *ACT7* promoter was overexpressed in Col-0 and *rmf1-1 rmf2-1* background. *RMF1-FLAG-OE* under control of the *ACT7* promoter was overexpressed in *rmf1-1* background. *RMF2-FLAG-OE* under control of the *ACT7* promoter was overexpressed in Col-0 background. *ASK1-MYC-OE* under control of the *ACT7* promoter was overexpressed in Col-0 background. *pRAD51::DMC1g/dmc1-2* and *pRAD51::DMC1g-6KR/dmc1-2* under control of the *RAD51* promoter was expressed in *dmc1-2* background. Plants were grown in a greenhouse under a 16-h-light and 8-h-dark photoperiod at 20 °C with 70% humidity. For media-cultured plants, seeds were sterilized with 75% ethanol and plated on 1/2 MS medium. Then the plants were moved to soil 7–10 days after germination on the medium and grown in the greenhouse. The WT tobacco plants were *Nicotiana benthamiana* grown under the same conditions as *Arabidopsis*.

## Plant morphology analysis

The plants and stems were photographed using a Canon digital camera SX20 IS (Canon). Pollen viability was detected by staining anthers with Alexander red at 65 °C for 1 h[62]. Tetrad-stage microspores were analyzed by staining microspores with toluidine blue at room temperature[63]. Images of stained pollen and microspores were obtained using a Zeiss Axio Scope A1 microscope (Zeiss).

## Chromosome morphology analysis

Chromosome spreading, fluorescence in situ hybridization (FISH) with a centromere probe, and immunofluorescence experiment were carried out as previously described[63]. Rabbit polyclonal antibodies of SYN1[45], ZYP1A[64], γH2AX[45], and DMC1[65] were used at a 1:200 dilution and the secondary antibody Alexa Fluor 555 Goat Anti-Rabbit IgG (H + L) (Invitrogen, A-21428) and Alexa Fluor 488 Goat anti-Rat IgG (H + L) (Invitrogen, A-11006) was used at a 1:1000 and 1:200 dilution in blocking buffer, respectively. Images were obtained using a Zeiss Axio Scope A1 microscope (Zeiss).

## Plasmid construction and plant transformation

To construct the vectors for yeast two-hybrid, pull-down, BiFC, co-IP, protein degradation, and ubiquitination assays, full-length CDS or genomic sequences of *RMF1/2*, *ASK1*, and *DMC1* were amplified by PCR using Phanta Super-Fidelity DNA polymerase (Vazyme Biotech, P515-03) with corresponding primers as described in Supplementary Data 4. All constructs were verified by DNA sequencing.

For *Arabidopsis DMC1-FLAG*, *RMF1/2-FLAG-OE*, *ASK1-MYC-OE* transgenic plants plasmid construction, the *DMC1*, *RMF1*, *RMF2*, and *ASK1* CDS sequences without stop codon were cloned and placed under the control of the *ACT7* promoter with modified pCAMBIA1306 plasmids (proACT7::3×FLAG, proACT7::5×MYC) using the One-step PCR Cloning Kit (Novoprotein, NR005-01B). For *Arabidopsis pRAD51-DMC1g/dmc1* and *pRAD51-DMC1g-6KR/dmc1* transgenic plants plasmid construction, the *DMC1* and *DMC1-6KR* genomic sequences (ATG to stop codon) were cloned and placed under the control of the *RAD51* promoter with a modified pCAMBIA1306 plasmid.

For constructions of transient expression in tobacco, the *DMC1/DMC1-6KR* CDS sequences for DMC1/DMC1-6KR-GFP and genomic sequences for DMC1/DMC1-6KR-FLAG, the *RMF1/2* CDS sequences for RMF1/2-FLAG or genomic sequences for RMF1/2-GFP, the *ASK1* CDS

sequence for ASK1-GFP and ASK1-MYC without stop codon were cloned and placed under the control of the 35S promoter with a modified pCAMBIA1306 plasmids (pro35S::3×FLAG, pro35S::5×MYC, pro35S::GFP).

For plant transformation, the constructs were transformed into agrobacterium GV3101 and transformed into *Arabidopsis* plants by floral dipping as previously reported[66]. The T1 plants were screened on 1/2 MS culture medium containing 25 mg/L hygromycin.

## Recombinant protein expression and purification

The RMF1/2 CDS sequences were cloned into pSUMO (on pET28a) and pMAL-c5X plasmids for constructions of SUMO-HIS-RMF1/2 and MBP-RMF1/2. The ASK1 CDS sequence was cloned into pGEX 4T-1 plasmid for constructions of GST-ASK1. The DMC1/DMC1-6KR CDS sequences were cloned into pSUMO (on pET28a) for constructions of SUMO-HIS-DMC1/DMC1-6KR. Then these plasmids were expressed in *E. coli* Rosetta (DE3) or BL21 (DE3) pLysS. GST, HIS, and MBP-fused proteins were induced with 0.02 mM IPTG for 12–16 h at 18 °C. Cells were disrupted with 30 min of sonication. Proteins were then purified by GST•Bind Resin (Merck, 70541), Ni-NTA HIS•Bind Resin (Merck, 70666) or Amylose Resin (NEB, E8021S), and eluted by 10 mM reduced glutathione (for GST-tagged recombinant protein), 350 mM imidazole (for HIS-tagged recombinant protein), 10 mM maltose (for MBP-tagged recombinant protein), respectively.

## Yeast two-hybrid assay

Target sequences from plasmid vectors were fused with the AD sequence in pGADT7 or the BD sequence in pGBKT7. The interaction between the prey vector (AD) and the bait vector (BD) or the transformants with either AD or BD were used as negative control. Transformants in two yeast strains were mated on YPDA plate for 24 h, and selected on SD/-Trp-Leu plates. Transformants were selected on SD/-His-Ade-Trp-Leu with X-α-Gal plates for protein interactions.

## In vitro pull-down assay

For each combination, 2 μg of GST, HIS, or MBP-tagged proteins were mixed in 400 μL binding buffer (50 mM Tris-HCl (pH 7.4), 150 mM NaCl, 25 mM imidazole, 0.5% NP-40) on a rotating wheel at 4 °C for 4 h, and then incubated with Ni-NTA HIS•Bind Resin at 4 °C for 2 h, followed by western blot analysis with anti-GST (Abmart, M20007, 1:2000 dilution), -HIS (Abmart, M30111, 1:2000 dilution), and -MBP (Proteintech, 15089-1-AP, 1:2000 dilution) antibodies, respectively.

## Split luciferase complementation imaging assay

ASK1 and DMC1 were fused to an N-terminal fragment of luciferase (nLUC), and RMF1 and RMF2 were fused to a C-terminal fragment of luciferase (cLUC) by One-step Cloning Kit. These vectors were transformed into agrobacterium GV3101. Transformants were collected when OD600 reached 1.0–1.5 and resuspended in suspension buffer (10 mM 4-morpholineethanesulfonic acid (MES), 100 μM acetosyringone (AS), 10 mM MgCl₂) to OD600 = 1.0. Suspensions were mixed in a 1:1 ratio for each pair of target proteins with additional 1/10 agrobacterium expressing P19 and then infiltrated into tobacco (*Nicotiana benthamiana*) leaves. After 36–48 h, leaves were sprayed with 250 μM luciferin with 1‰ Triton X-100 and monitored using a living plant imaging system NightSHADE LB 985 (Berthold Technology) with indiGo software (v2.0.5.0).

## Bimolecular fluorescent complementation (BiFC) assay

The CDS sequences of RMF1/2, ASK1, and DMC1 were cloned into BiFC plasmid pXY103/104/105/106 by One-step Cloning Kit and transformed into agrobacterium GV3101. Transformants were collected at an OD600 of 2.0 and resuspended in suspension buffer (10 mM 4-morpholineethanesulfonic acid (MES), 10 mM MgCl₂, 100 μM acetosyringone (AS)) to OD600 = 1.0. Suspensions were mixed in a 1:1 ratio

for each combination and infiltrated into tobacco leaves. After 36–48 h, leaves were collected and observed using a laser confocal microscope FV3000 (Olympus).

### Co-immunoprecipitation (Co-IP) assay

Tobacco leaves co-infiltrated with 35S::RMF1/2-FLAG and 35S::ASK1-MYC or 35S::RMF1/2-GFP and 35S::DMC1-FLAG were ground to a fine powder in liquid nitrogen and homogenized in protein lysis buffer (50 mM Tris-HCl (pH 8.0), 100 mM NaCl, 10% glycerol, 0.5% NP-40, 1% cocktail proteinase inhibitor). The supernatants were incubated at 4 °C for 1 h, and cell debris was removed by centrifugation at $21,130 \times g$ for 10 min. After preclearing with anti-FLAG magnetic beads (Bimake, B26102) or anti-GFP magnetic beads (KT Life Technology, KTSM1334), the lysates were incubated with corresponding beads at 4 °C overnight. The immunoprecipitation complexes were washed three times with 1 ml protein lysis buffer. Proteins retained on beads were separated by SDS-PAGE and detected using anti-FLAG (GNI, GNI4110-FG, 1:2000 dilution), -MYC (Sigma-Aldrich, 05-724-25UG, 1:2000 dilution), or -GFP (GNI, GNI4110-GP, 1:2000 dilution) antibodies.

### Western blot

Total protein was extracted using protein lysis buffer with ground central inflorescences. 5× SDS loading buffer was added to the supernatants and the lysates were boiled at 95 °C for 6 min. Samples were separated by SDS-PAGE using 10% acrylamide gels and electro-blotted to nitrocellulose (NC) membranes (Abm, B500) and incubated in anti-FLAG, -MYC, -GFP, -UBQ11 (Agrisera, AS08 307A, 1:5000 dilution), -DMC1 (1:500 dilution)[65], -β-Tubulin (Abmart, M20005, 1:2000 dilution), and -HSC70 (ENZO, ADI-SPA-818-F, 1:2000 dilution) antibodies. Proteins were detected using corresponding antibodies. Bands were visualized with a Clinx-3400 chemiluminescence imaging system. Goat anti-mouse IgG HRP-conjugated antibody (GNI, GNI9310-M, 1:2000 dilution) and goat anti-rabbit IgG HRP-conjugated antibody (GNI, GNI9310-R, 1:2000 dilution) were used as the secondary antibody. Source data of the most important blots are provided.

### Quantitative RT-PCR (qRT-PCR)

Total RNAs were extracted from central inflorescences using Trizol reagent (Invitrogen, 15596026), and reverse transcription was performed with 1 μg total RNA (PrimeScriptTM II reverse transcriptase, Takara, RR047A). qRT-PCR was performed using qPCR SYBR Green Master Mix (YEASEN, 11198ES08) with Bio-Rad CFX96 Touch Thermocycler (Bio-Rad) and data were collected and analyzed with Bio-Rad CFX Manager (v3.1). Each transgenic plant includes three independent lines and each qRT-PCR experiment included three technical replicates. TIP41-like was used as the reference gene, and gene expression level was calculated by the $2^{-\Delta\Delta Ct}$ method as previously reported[67] then compared with Col-0. The statistical significance (p values) of differences in gene expression levels between samples was analyzed using a two-tailed Student's t-test. The position of qRT-PCR primers are indicated on corresponding gene schematic diagrams. The qRT-PCR primers are described in Supplementary Data 4.

### In vivo ubiquitination assay

Total proteins from central inflorescences of ProACT7::DMC1-FLAG transgenic plants or tobacco leaves infiltrated with 35S::DMC1-GFP were extracted as described above. The lysates were incubated with anti-FLAG magnetic beads, anti-GFP magnetic beads, or TUBE2 agarose beads (Life Sensors, UM402) at 4 °C for 4 h. Immunoprecipitation complexes were washed three times with 1 ml protein lysis buffer. 2× non-reducing SDS loading buffer with 100 mM DTT were added, and boiled at 95 °C for 6 min. DMC1 ubiquitination levels were detected by Western blot analysis with anti-FLAG, -GFP, -UBQ11, and -DMC1 antibodies.

### In vitro ubiquitination assay

Recombinant SUMO-HIS-DMC1 was used as substrate. RMF1/2-containing E3 ubiquitin ligases were immunoprecipitated from inflorescences of RMF1/2-FLAG-OE transgenic plants with anti-FLAG magnetic beads. 300 ng E1 (yeast UBE1, Boston Biochem, E-300), 400 ng E2 (human UbcH5c, Boston Biochem, E2-625), 2 μg purified HIS-ubiquitin, 50 μM MG-132 and 1× reaction buffer (50 mM Tris-HCl (pH 8.0), 10 mM MgCl$_2$, 1 mM ATP) were included for the reaction. After incubating at 37 °C for 1 h, the reaction was stopped by adding 2× non-reducing SDS loading buffer with 100 mM DTT, and the mix was boiled at 95 °C for 6 min. DMC1 ubiquitination levels were detected by western blot analysis with anti-DMC1 and anti-UBQ11 antibodies.

### In vivo MG-132 suppress substrate degradation assay

In vivo MG-132 suppressing substrate degradation assay was performed as previously reported[68]. Agrobacterium strains carrying 35S::DMC1-GFP or 35S::DMC1-6KR and 35S::GFP as an internal control were co-infiltrated into tobacco leaves. 50 μM MG-132 as experimental group and 10 mM MgCl$_2$ diluent as control group were infiltrated into the same tobacco leaves for 12 h before sample collection. Samples were harvested at 2 days after infiltration and analyzed by western blot analysis using anti-GFP and -HSC70 antibodies.

### In vivo E3 ligase-promoted substrate degradation assay

The in vivo E3 ligase-mediated substrate degradation assay was performed as previously reported[68]. Agrobacterium strains carrying 35S::DMC1-GFP or 35S::DMC1-FLAG and 35S::GFP as an internal control were co-infiltrated with different ratios of 35S::RMF1/2-FLAG or 35S::ASK1-GFP into tobacco leaves. Samples were harvested at 2 days after infiltration and analyzed by western blot analysis using anti-FLAG, -GFP, and -HSC70 antibodies.

### Semi-in vivo E3 ligase-promoted substrate degradation assay

The semi-in vivo E3 ligase-mediated substrate degradation assay was performed as previously reported[68]. RMF1/2 and DMC1 were expressed separately by infiltrating agrobacterium strains carrying 35S::DMC1-FLAG and 35S::RMF1/2-GFP into different tobacco leaves. Infiltrated samples and untreated leaves as mock were harvested at 2 days after infiltration and total proteins were extracted as described above. 10 mM ATP was added to maintain the proteasome function. DMC1 extract was mixed with extract of RMF1/2 or mock in a 1:1 ratio and incubated at room temperature on a tottering mixer and removed equally at separate time points. The reaction was stopped by addition of 2× SDS loading buffer and boiled at 95 °C for 6 min and analyzed by western blot analysis using anti-FLAG, -GFP and -HSC70 antibodies.

### Cell-free degradation assay

Central inflorescences of Col-0 and rmf1-1 rmf2-1 were ground to a fine powder in liquid nitrogen, then 1 mL protein lysis buffer was added to extract the total protein at 4 °C for 1 h. The supernatant was collected by centrifugation at $21,130 \times g$ for 10 min, and adjusted to equal concentrations with protein lysis buffer. 150 μM cycloheximide (CHX) were added to inhibit protein biosynthesis and 10 mM ATP was added to maintain the activity of proteasome. 50 μM MG-132 was added for MG-132 treatment. 1 μg purified recombinant SUMO-HIS-DMC1 or SUMO-HIS-DMC1-6KR proteins were incubated with 5 mg Col-0 or rmf1-1 rmf2-1 supernatants at room temperature, and aliquots were removed at separate time points. The reaction was stopped by adding 2× SDS-PAGE loading buffer and boiled at 95 °C for 6 min. The samples were resolved by 10% SDS-PAGE. Proteins were detected using an anti-HIS antibody.

### Phylogenetic analysis of RMF1/2 homologs

Homologs of Arabidopsis RMF1 and RMF2 were identified using HMMER3[69] to search against a protein database from selected plant

species from Phytozome v13 (https://phytozome-next.jgi.doe.gov). The initial phylogenetic tree was constructed using IQ-Tree[70]. RMF1/2 and their homologs sharing the same latest common ancestor in land plants were selected according to it. A final tree was also constructed using IQ-Tree with the JTT + G4 mode and parameters "--altr 1000 -B 1000" and visualized by FigTree.

## LC-MS and data processing

The purified protein samples were reduced with 10 mM DTT at 37 °C for 45 min and then alkylated with 100 mM acrylamide at 25 °C for 1 h. The FASP method was adopted for protein digestion. The protein samples were transferred to Microcon PL-10 filters and centrifuge at 13,800 × g for three-time buffer displacement, and then dissolved with digestion buffer (50 mM Tris-HCl, pH 8.0). Trypsin was added at a ratio of enzyme/protein (w/w) as 1:50 and the solutions were incubated at 37 °C for overnight digestion. After digestion, the solution was collected by centrifugation and the filter was washed twice with 10% ACN. Finally, the filtrates were pooled and vacuum-dried in a SpeedVac.

LC-MS/MS analysis was performed using a nanoflow EASY-nLC 1200 system (Thermo Fisher Scientific) coupled to an Orbitrap Fusion Lumos mass spectrometer (Thermo Fisher Scientific). Samples were loaded and analyzed on an in-house packed C18 columns (75 μm i.d. × -20 cm; 1.9 μm, ReproSil-Pur 120 C18-AQ, Dr. Maisch GmbH). The mobile phases consisted of solvent A (0.1% formic acid) and solvent B (0.1% formic acid in 80% ACN). The peptides were eluted using the following gradients: 5–8% B in 3 min, 8–44% B in 45 min, 44–100% B in 2 min, and 100% B for 10 min at a flow rate of 200 nL/min. Data acquisition mode was set to obtain one MS scan followed by sequential HCD-MS/MS acquisitions with a cycle time of 3 s. All spectral data were detected by Orbitrap mass analyzer. For the MS scans, the mass range was set as 350–1600 at a resolution of 60,000, and the automatic gain control (AGC) target was set as standard. For the MS/MS scans, the resolution was set as 15,000, the precursor isolation window was 1.6 $m/z$, the dynamic exclusion was 30 s with a precursor mass tolerance of 10 ppm.

The raw data were analyzed using Proteome Discoverer (v1.4, Thermo Fisher Scientific) using an in-house Mascot Server (v2.7, Matrix Science). Data were searched using the following parameters: the protein sequences downloaded from TAIR were set as database; trypsin/P as the enzyme; up to four missed cleavage sites were allowed; 10 ppm mass tolerance for MS and 0.05 Da for MS/MS fragment ions; propionamidation on cysteine as static modification, oxidation on methionine and ubiquitination on lysine as variable modification. The incorporated Percolator in Proteome Discoverer was used to validate the search results and only the hits with FDR ≤ 0.01 were used for analysis.

## Prediction of DMC1 and DMC1-6KR protein structure using AlphaFold protein structure database

The online AlphaFold protein structure database (https://alphafold. ebi.ac.uk/) and the online AlphaFold2 protein structure prediction database (https://colab.research.google.com/github/sokrypton/ ColabFold/blob/main/AlphaFold2.ipynb)[50] were used to predict the protein structure of DMC1 and DMC1-6KR. The final protein structures were visualized by PyMOL.

## Statistics

The online PANTHER database (http://www.pantherdb.org/) was used for GO term analysis and statistics were calculated by $p$ values correction. R (v3.4.2) was used for drawing the heat map of expression level in meiocyte and leaf of 62 meiocyte-specific or preferential expressed F-box genes in *Arabidopsis*. MEGA (v6.0) was used for alinement of DMC1 amino acids sequences in multiple species, and the alinement was visualized by DNAMAN (v6.0.3). Excel 2019 (Microsoft) and GraphPad Prism 7 were used for calculating the mean and

standard deviation of pollen numbers, tetrad numbers, γH2AX foci numbers, DMC1 foci numbers, and gene expression levels. Significance was tested using two-tailed Student's $t$-test or one-way ANOVA test with 95% confidence interval. Photoshop CS6 software was used for resizing and adjusting the images. ImageJ software was used for all image quantifications to obtain parameters of band density from control and experimental groups.

## Accession numbers

*Arabidopsis* Genome Initiative (AGI) gene identifiers used in this study are: AT4G29420, AT2G29610, AT3G10430, AT2G17830, AT5G42460, *RMF1* (AT3G61730), *RMF2* (AT5G36000), *ASK1* (AT1G75950), *DMC1* (At3g22880), *SPO11-1* (AT3G13170), *RAD51* (AT5G20850), *RAD51C* (AT2G45280), *XRCC3* (AT5G57450), *MRE11* (AT5G54260).

## Reporting summary

Further information on research design is available in the Nature Portfolio Reporting Summary linked to this article.

## Data availability

The amino acid, CDS and genomic sequences of RMF1, RMF2, ASK1, DMC1 and RAD51 are available at TAIR (https://www.arabidopsis.org/) under the accession numbers *AT3G61730* (*RMF1*) [https://www. arabidopsis.org/servlets/TairObject?id=36024&type=locus], *AT5G36000* (*RMF2*) [https://www.arabidopsis.org/servlets/TairObject? id=133155&type=locus], *AT1G75950* (*ASK1*) [(https://www.arabidopsis. org/servlets/TairObject?id=137570&type=locus], *At3g22880* (*DMC1*) [https://www.arabidopsis.org/servlets/TairObject?id=37395&type= locus], *AT5G20850* (*RAD51*) [https://www.arabidopsis.org/servlets/ TairObject?id=130976&type=locus], respectively. Homologs of *Arabidopsis* RMF1 and RMF2 for phylogenetic analysis were identified using HMMER3 to search against a protein database from selected plant species from Phytozome v13 (https://phytozome-next.jgi.doe.gov). The predicted protein structure of DMC1 and DMC1-6KR were obtained from the online AlphaFold protein structure database (https://alphafold.ebi. ac.uk/) and the online AlphaFold2 protein structure prediction database (https://colab.research.google.com/github/sokrypton/ColabFold/blob/ main/AlphaFold2.ipynb). The mass spectrometry proteomics data generated in this study have been deposited in the ProteomeXchange under accession code PXD038116 and PXD038126. Source data are provided with this paper.

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

## Acknowledgements

We greatly thank Prof. Binglian Zheng from Fudan University for providing the pMAL-c5X, JW771, and JW772 plasmids and pET-28a-ubiquitin plasmid, Prof. Jinbiao Ma from Fudan University for providing the pSUMO (on pET-28a) and GEX-4T-1 plasmids. This work was supported by grants to Y.W. from the National Natural Science Foundation of China (31925005), Double first-class discipline promotion project of SCAU (2023B10564004), and Guangdong Laboratory for Lingnan Modern Agriculture (NG2022002).

## Author contributions

W.X., Y.H., and Y.W. conceptualized the study. W.X. performed the experiments with help from Y.Y., J.J. generated the *rmf1-2* and *rmf2-2* mutants, Z.W. and X.Z. performed the mass spectrometry assay to detected ubiquitination sites of DMC1. C.Y. drew the phylogenetic tree. H.M. provided the *ask1-1* mutant. W.X., C.Y., G.P.C., and Y.W. interpreted date, wrote, and edited the manuscript with comments from all authors.

## Competing interests

The authors declare no competing interests.
