## [Peer Review File · Nature Communications]

SCFRMF Mediates Degradation of the Meiosis-Specific Recombinase DMC1REVIEWER COMMENTS

Reviewer #1 (Remarks to the Author):

Dear Editor(s) of Nature Communications,

The manuscript with the title "SCFRMF-mediated degradation of the recombinase DMC1 ensures meiotic recombination", submitted by the authors Wanyue Xu, Yue Yu, Juli Jing, Zhen Wu, Xumin Zhang, Chenjiang You, Hong Ma, Gregory P. Copenhaver, Yan He and Yingxiang Wang contains a very interesting, timely and mostly well-conducted study of a crucial and understudied biological aspect: protein stability regulation during meiosis. Moreover, the findings relate to a conserved protein factor (DMC1) that is key to meiotic recombination and inter-homologous DNA repair and thus address a biological problem that is relevant beyond the field of plant sciences. The study is technically sophisticated and employs multiple different approaches to address the research questions from different experimental angles. Most (see points of criticism below) experiments are elegant and convincing and the overall conclusions, if supported by all experiments, are/would be very relevant and important for the understanding of meiosis, not only in plants. There are a couple of issues that need to be addressed. See below:

Major issues:

Please indicate if the many transgenic lines / transient transfections in tobacco expressing (in some instances tagged) variants of RMF1, RMF2, ASK1, DMC1 were actually functional or not. While this is most likely not relevant for some experiments it for sure is for others. If functional complementation tests were not done, please indicate the fact and argue why in the particular cases it was not deemed necessary and why the conclusions from the corresponding experiments are still considered valid. As "devil's advocate" one could ask if only DMC1 proteins that are aberrantly folded and not functional (potentially the tagged versions) are targeted for degradation. I am sure you have good arguments against this concern and it would be great to read them in the discussion.

The anti-DMC1 antibody was first introduced in 2019 by the same group (Wang et al), yet it turns out that neither in the former publication nor in this manuscript a proper control for its specificity is given. If I understand correctly then in Figure 5F a *dmc1* mutant is employed and does not give a signal using the AB (which is great!). I guess, given the experimental sets, it would be a minor effort to provide an IF image of *dmc1* mutants stained with this AB and also a Western blot with heterologously expressed DMC1 and RAD51 using the AB and demonstrating specificity.

Comments below regarding Figure 5C.

Comment below regarding Suppl. Fig. 6 G-J.

Minor issues:

Abstract:

Line 24/25: pl revise sentence.

Introduction:

Line 41: "DNA" double strand breaks...pl insert.

Line 45/46: pl revise sentence: homologs are not paired at the stage of invasion...

Line 47:was found the be defective....pl rephrase

Line 49: the number of RecA homologs in eukaryotes is actually higher, as also the plastid and

mitochondria proteins could be considered.... Pl change number or writing...

Line 58: Mei5/Sae3 is one of the accessory complexes, yet it is not conserved in plants; pl also discuss and include the conserved (present and crucial in plants) accessory complex Mnd1/Hop2.

Line 58/59: "... and its dissociation from dsDNA relies on the Rdh54/Tid1 translocase and its paralog Rad54 in an ATP-dependent manner...." Pl rephrase.

Line 72: "In budding yeast....": Logic break; why these E3s? Connection sentence missing...

Lines 88/89: Check logic; do u really want to say this? Maybe something like this: ...the targets that are controlled by the ub prot pathway are largely uncharacterized.....?

Results and Figures:

Suppl. Figure 1B: how was the expression test done? Primers against DMC1 in general or the transgenic part (e.g the FLAG part)?.... the primer set P5 does not indicate this; Please extend the table with a clearer description, indicate in the legend and provide a schematic drawing indicating the positions of the used primer (pl do this for all primer sets and for all (qRT) PCR experiments).

Line 105: "1009 proteins": pl provide a separate sheet in the file: provide the hits after background removal and indicate those with the GO-term "ubiquitin proteasome pathway", as a service to the reader.

Line 111: I could not find the corresponding construct in M&M section and also not the transient transfections with this construct....pl check and complete if needed. This relates to the general aspect mentioned at various occasions: pl make sure that constructs are better described and cross-referenced in the M&M section with respect to the individual experiments.

Suppl. Fig. 2 and 3: in the pptx there is a mis-labelling of the figure # in the right lower corner.

Line 160: "sterile pollen" does not exist....pl correct.

Line 196: pl tame down sentence:...no proof at that stage of manuscript; rather be specific and say that RMF1 RMF2 act downstream of meiotic DSB formation as does DMC1 and that they do not have an additive effect suggesting that they could....etc..

Suppl. Figure 5C: Pl. extend figure legend and provide info which heterologous system for expression was used (pl check entire manuscript and clarify all such cases). Moreover, the labelling of the second panel confusing, pl fix.

Figure 4A:

General comment: figure and text flow are not always congruent. Pl fix for entire manuscript (e.g.: Fig 3E/D are discussed after introducing Figure 4A).

Line 233: "to test this hypothesis..."; pl write "to corroborate this hypothesis...."

Line 252: sentence confusing: pl rephrase...(you used DMC1 as a substrate, heterologously expressed and purified from E. coli.....this is what you mean...right?)

Line 254: I guess the term "purified" is not appropriate in this context. Use e.g. "captured"

Figure 2:

C-E: Pictures of anthers not as nice as the one in the Suppl. Figure; pl provide better pictures of

anthers better demonstrating the full viability.

Legend of Figure 2:

Line 870 – “fertile siliques” do not exist...pl rephrase;

Line 871 – “sterile siliques” neither...pl rephrase.

Line 886 – “have defective” ...pl correct.

Line 887 – delete the term “recombination”...this is not assayed here.

Figure 5C:

Controls are missing: e.g. probing the blot with an anti-Ub antibody; e.g. loading E1 and E2 alone in separate lanes.

The bands that are indicated to represent polyub-DMC1 seem distinct and it is unlikely they represent the 8kDa increments for each single Ub moiety that is additionally attached....

Line 287:

The MS experiment is not described in the M&M section. Pl provide details: endogenous protein; OE lines; tobacco experiments; controls....etc....

Line 298: the statement that “...suggesting that the mutations do not cause gross structural changes and may not impede recombinase activity ...” has to be deleted as there is no experimental support.

Line 299: please state that it is “nuclear localisation” in tobacco cells....

Figure 5D:

It is not clear which DMC1-OE line was used here...pl specify.

Pl provide entire blot for both ABs...the image is unclear.

Figure 5F:

Which of the OE plants (shown and characterized in the supplements....plant 1-3) are used here?

Again, pl provide clearer info if cDNA or genomic versions used and if functional complementation tests were performed.

Figure 6B last lane: double band for DMC1-6KR? Please comment. Furthermore, it appears that the labelling of this lane is not correct. Pl check.

Figure 6F: is unclear where the tagged DMC1 protein comes from (expression system, purification).

Suppl. Fig. 6 G-J: it is unusual to present new data in the discussion section. Move to the results part. Moreover, this is most likely one of the most problematic parts of the manuscript. The over-expresser lines are in principle very interesting as they could settle the question if meiotic inter-homologue DNA repair is indeed a fine-tuned aspect of DMC1 protein abundance. Yet the way the experiments have been performed and/or are presented does not allow final conclusions: Are the presented lines based on cDNA or on genomic DNA....please clarify; It appears that the OE lines are FLAG tagged...pl indicate within the figure. Most importantly: are the over-expressed proteins functional: do they complement the dmc1 mutant phenotype? This would be crucial, as so far, no tagged and functional version of DMC1 is known in any organism. While the conclusions the authors are drawing could be true (provide all controls would be in place) it could also be that they introduced dominant-negative versions of tagged DMC1, which would then demand a different interpretation. Pl clarify/adapt/explain.

Material and Methods:

Please provide a clearly depiction which constructs make use of genomic versions of a gene, which of cDNA; provide more indicators which construct was used in which experiment (cross-references to Figures/Suppl. Figures). As mentioned at various occasions: make sure to indicate if a construct was tested for functional complementation or not; provide explanations why conclusions may still be valid

from given experiment if not tested. Point this out redundantly: in the text body and the M&M section.

Line 630 ff: details missing regarding elution and further purification; validation of protein ID, purity and yield missing; pl provide.

Discussion:

Line 330: either references or the sentence are wrong; the references relate to RAD51 being ubiquitinated....pl fix.

Lines 355 ff:

"In this study, our higher-order mutants analysis shows that the absence of RAD51 and RMF1/2 and/or ASK1 dramatically decreases meiotic chromosome fragmentation as long as DMC1 is active."...pl rephrase sentence....it is confusing.

Line 358: pl use small letters and italics for mutants...otherwise it is confusing.

Please include the following in your discussion:

Pl discuss why DMC1 was not found in the study for ASK1 targets: Lu et al 2016 PMID: 26940208

Pl discuss study of PMID: 14756314 and possible redundancy between ASK1 and ASK2

Reviewer #2 (Remarks to the Author):

NCOMMS-22-45529-T

Xu et al.

The manuscript by Xu et al. described the ubiquitination of a meiosis-specific recombinase, DMC1, by the SCF E3 ubiquitin ligase complex of the SCF with RMF1/2 as an F-box protein in *Arabidopsis thaliana*. The authors showed that DMC1 binds to RMF-1/2 in both in vivo and in vitro and identified a possible ubiquitination of DMC1 using the expression in tobacco leaf and by in vitro reconstitution. Importantly, the *rmf-1 rmf-2* double mutants and a mutant in the ASK-1 encoding a SKIP1-like protein showed meiotic defects similar to the *dmc1* mutant. In these mutants, the level of DMC1 protein is increased compared to the wild-type control. The same is true for DMC1 protein (DMC1-6KR) with amino-acid substitutions for lysine for possible ubiquitination. This study reveals a new type of regulation of DMC1-mediated meiotic interhomolog recombination, that is associated with ubiquitin-mediated protein degradation. The experiments have been done in a good shape and most of the results are convincing and provide new insight. However, how increased levels of DMC1 in the *asf-1* and *rmf-1 rmf-2* double mutants contribute to meiotic defects is not clearly documented. And the role of ASK1, RMF1/2-dependent removal of DMC1 from meiotic chromosomes has not been analyzed in detail. As shown below, these should be addressed experimentally and discussed well. And the points listed below should be evaluated prior to publication.

Major points:

1. In Figures 1 and 6, the authors showed in vivo ubiquitination of DMC1 in tobacco leaves. Interestingly, the main band of both DMC1-GFP or DMC1-FLAG can be detected by not only anti-DMC1 but also anti-UBQ11. Usually, shifted or smear bands of DMC1, not the main band would be reacted with anti-UBQ11. It is strange that DMC1 bands reacted with anti-UBI11 do not show any upper band shift relative to DMC1 major band detected with either GFP or DMC1 antibodies. Does this mean that the major DMC1 band is a constitutively mono-ubiquitinated DMC1? In the same line, TUBE2-purified DMC1, which should be ubiquitinated, also shows a sharp band in Figures 1D, E, and 6C, suggesting that the major DMC1 band contains the ubiquitin-binding ability reacted with TUBE2. It is essential to explain these strange results.
2. The univalent formation in the *ask-1*, *rmf-1 rmf-2*, *ask-1 rad51* and *rmf-1 rmf-2 rad51* mutants

(Figure 3A, D) suggests the DSBs are repaired by inter-sister recombination. Thus, these mutants with wild-type or more DMC1 focus number (Figures 5A and B) can dismantle DMC1 from the chromosomes for the completion of the recombination without ASK1 or RMF1/2. On the other hand, the authors claim the role of ASK1-RMF1/2 removal of DMC1 in wild-type situations, but not in the mutants (Figure 7). This discrepancy should be discussed well. In the line, it is very important to check the DMC1 focus formation (with the quantification) in the late stages of meiotic prophase I such as diakinesis in the *ask-1*, *rmf-1 rmf-2*, *ask-1 rad51*, and *rmf-1 rmf-2 rad51* mutants. If ASK1-RMF1/2 can remove DMC1 filaments for proper interhomolog recombination (Figure 7), one can expect the accumulation of DMC1 foci on chromosomes in diakinesis.

3. The univalent formation in the *ask-1 rad51* and *rmf-1 rmf-2 rad51* mutants also suggests that, in these mutants, DMC1 cannot carry out interhomolog recombination, but rather can do intersister recombination. This point should be discussed. And, importantly, to extend the idea, the chromosome synapsis should be studied by the staining for ZYP1A in these mutants.

4. Figure 6, the treatment with MG132 stabilizes the steady-state level of wild-type DMC1 in tobacco leaves (increased 10-fold or ~4-fold in Figure 5E). In the tobacco leaf system, what happens to the expression of RMF1/2 and ASK1? Indeed, does the ubiquitination and/or degradation of DMC1 depend on both RMF1/2 and ASK1?

5. The suppression of the *rad51* mutant by the SCF mutants is similar to the results in the recent publication showing that a mutation in the SMC5/6 genes suppresses the *rad51*'s meiotic defects including DMC1 focus formation (Chen, *Plant Cell*, 33, 2869-, 2021). The relationship of ASK1-RMF1/2 with SMC5/6 should be experimentally addressed or discussed.

Minor points:

1. Figure 4E and 4F: These results should be in Supplement without the analysis of the domains for in vivo or in vitro ubiquitination.

2. Figures 5A and 5B: Please add DMC1 focus counting in pachytene or pachytene-like stages in each strain to validate the claim (lines 236-238).

3. Figure 5D bottom panel: Please show a full blot with a DMC1-FLAG band like in Figures 1D and 1E.

4. Lines 179-181: *ask-1 spo11* and *rmf-1 rmf-2 spo11* are clearly different from *spo11* in anaphase I chromosome morphology. The *spo11* shows a more compacted univalent while the *ask-1 spo11* and *rmf-1 rmf-2 spo11* mutants show an elongated or V-shaped univalent. This suggests that the *ask-1* and *rmf-1 rmf-2* mutants show Spo11-independent meiotic defects (opposed to the claim shown in lines 182-184). This point should be clarified, and the authors need a quantitative analysis of chromosome defects.

5. Line 195, meiotic recombination defects: Figure 3C shows only chromosome defects, not recombination defects. The authors need to be careful about the conclusion from the results.

6. Line 199: The authors pointed out ASK2 in addition to ASK1. Since *ask-1* single mutant showed meiotic defects. The authors need more explanation on the ASK2 more in its function.

7. Line 268: What is the reason why the authors used Mg²⁺ as a control of MG-132?

8. Line 271-273: What is the phenotype of RMF1/2 or ASK1 OE plant? Given the reduced DMC1 levels, these mutants should show a meiotic recombination defect like the *dmc1*. DMC1 focus and chromosomal abnormality should be presented.

Reviewer #3 (Remarks to the Author):

The manuscript "SCFRMF-mediated degradation of the recombinase DMC1 ensures meiotic recombination" by Wang and co-authors presents the results of an in-depth study of Arabidopsis DMC1 and more specifically how its regulated degradation is essential for homologous recombination during meiosis. The authors describe an elegant series of experiments that result in the identification of the redundant F-box genes RMF1/2 and show that RMF1/2 interact with the Skp1 ortholog ASK1 to form a ubiquitin ligase complex SCFRMF1/2 that is responsible for DMC1 ubiquitination and its subsequent degradation by the 26S proteasome.

The results of the work are highly original further our understanding of how DMC1 is regulated and make important contributions to the field of meiotic homologous recombination, and meiosis in general. This is the first evidence that DMC1 is ubiquitinated in plants and I believe the first instance of results describing a detailed mechanistic picture of how specifically DCM1, a key meiotic recombinase that is broadly conserved in eukaryotes is regulated in any organism.

The manuscript is generally well-written, and the data presented is extensive and of high quality. The conclusions drawn are significant and are supported by the result presented. In many instances the authors present the results of multiple experiments in support of a particular conclusion. My only suggestion is that the manuscript be given a careful proof-reading as there are many small grammatical errors sprinkled throughout the manuscript.

Dear Nature Communications Editors and Reviewers,

Thank you very much for your helpful comments and suggestions about our manuscript entitled “SCF^{RMF} Mediates Degradation of the Meiosis-Specific Recombinase DMC1” (Tracking NO. 22-45529-T). We are pleased to resubmit our revised manuscript for your consideration. Below, please find our point-by-point response (questions in black and responses in blue).

Reviewer #1:

Dear Editor(s) of Nature Communications,

The manuscript with the title “SCF^{RMF}-mediated degradation of the recombinase DMC1 ensures meiotic recombination”, submitted by the authors Wanyue Xu, Yue Yu, Juli Jing, Zhen Wu, Xumin Zhang, Chenjiang You, Hong Ma, Gregory P. Copenhaver, Yan He and Yingxiang Wang contains a very interesting, timely and mostly well-conducted study of a crucial and understudied biological aspect: protein stability regulation during meiosis. Moreover, the findings relate to a conserved protein factor (DMC1) that is key to meiotic recombination and inter-homologous DNA repair and thus address a biological problem that is relevant beyond the field of plant sciences. The study is technically sophisticated and employs multiple different approaches to address the research questions from different experimental angles. Most (see points of criticism below) experiments are elegant and convincing and the overall conclusions, if supported by all experiments, are/would be very relevant and important for the understanding of meiosis, not only in plants. There are a couple of issues that need to be addressed. See below:

Response: We appreciate the reviewer’s positive assessment of our manuscript and have carefully considered their comments and revised the manuscript accordingly.

Major issues:

Please indicate if the many transgenic lines/transient transfections in tobacco expressing (in some instances tagged) variants of RMF1, RMF2, ASK1, DMC1 were actually functional or not. While this is most likely not relevant for some experiments it for sure is for others. If functional complementation tests were not done, please indicate the fact and argue why in the particular cases it was not deemed necessary and why the conclusions from the corresponding experiments are still considered valid. As “devil’s advocate” one could ask if only DMC1 proteins that are aberrantly folded and not functional (potentially the tagged versions) are targeted for degradation. I am sure you have good arguments against this concern and it would be great to read them in the discussion.

Response: We agree with the reviewer that some transient transfections in tobacco expressing tagged variants of RMF1, RMF2, ASK1, DMC1 were not tested for functional complementation. As pointed out by the reviewer, the tagged proteins in transgenic plants may have aberrant folding which may in turn influence degradation, we functionally tested the constructs used for overexpressing DMC1 in WT and found that they did not complement (*pDMC1-DMC1g-FLAG/Col-0* and *pHTR2-DMC1g-FLAG/Col-0*, the original Supplementary Fig. 6d-j). As a result, we have removed the original Supplementary Figure 6 g-j. However, we have added new data using un-tagged versions of DMC1 - please see the response to major issue #4 in

page 7-8 (revised Supplemental Figure 10) below for detail. Some of the transgenic lines we used were not tested for functional complementation, but an anti-DMC1 antibody was used to measure DMC1 protein levels in multiple mutants including *rmf1 rmf2* and *ask1* (the revised Figure 6C), and these results were consistent with the results using tagged proteins, supporting the idea that ASK1-RMF1/2 are responsible for DMC1's degradation.

We have acknowledged this caveat in the text (line 309-313 in page 11-12) by adding the following statement: The tagged versions of proteins may not be able to complement full biological function of their wildtype counterparts, but taken as a whole, the experiments using tagged proteins are consistent with the finding that ASK1-RMF1/2 facilitate DMC1's degradation.

The anti-DMC1 antibody was first introduced in 2019 by the same group (Wang et al), yet it turns out that neither in the former publication nor in this manuscript a proper control for its specificity is given. If I understand correctly then in Figure 5F a *dmc1* mutant is employed and does not give a signal using the AB (which is great!). I guess, given the experimental sets, it would be a minor effort to provide an IF image of *dmc1* mutants stained with this AB and also a Western blot with heterologously expressed DMC1 and RAD51 using the AB and demonstrating specificity.

Response: As suggested by the reviewer, we have conducted an IF experiment of *dmc1* mutants stained with anti-DMC1 antibody and did not find any signals of DMC1 in *dmc1-2* mutant. We have included the image in the revised Fig.5c (the second lane). We also used Western blotting of recombinant SUMO-HIS-DMC1 and -RAD51 with anti-HIS and anti-DMC1 antibodies and found that the anti-DMC1 antibody recognizes DMC1 but not RAD51 (see below and also Supplementary Fig. 1d in the revised manuscript), which demonstrates the specificity of the anti-DMC1 antibody.

Specificity of anti-DMC1 antibody examination using recombinant SUMO-HIS-DMC1 and SUMO-HIS-RAD51 heterologously expressed in *E.coli*.

Comments below regarding Figure 5C.

Response: Please see the response in page 5-6 below.

Comment below regarding Suppl. Fig. 6 G-J.

Response: Please see the response in page 8 below.

Minor issues:

Abstract:

Line 24/25: p1 revise sentence.

Response: We have revised the sentence to read “Here, we show that the degradation of *Arabidopsis* DMC1 by the 26S proteasome depends on ubiquitination mediated by the F-box proteins RMF1/2”.

Introduction:

Line 41: “DNA” double strand breaks...pl insert.

Response: We have inserted “DNA” as suggested.

Line 45/46: pl revise sentence: homologs are not paired at the stage of invasion...

Response: We have revised the sentence as “which invade homologs to form a recombination intermediate called a D-Loop”.

Line 47:was found the be defective....pl rephrase

Response: We have revised the sentence as “The first *recA* mutant was isolated in *E.coli* (*Escherichia coli*) and was found to be defective in recombination and DNA damage repair”.

Line 49: the number of RecA homologs in eukaryotes is actually higher, as also the plastid and mitochondria proteins could be considered.... Pl change number or writing...

Response: We have revised the sentence as “Eukaryotes contain multiple RecA family proteins”.

Line 58: Mei5/Sae3 is one of the accessory complexes, yet it is not conserved in plants; pl also discuss and include the conserved (present and crucial in plants) accessory complex Mnd1/Hop2.

Response: We have revised the sentence as “The assembly of Dmc1 on RPA-coated ssDNA is mediated by Mei5-Sae3 in budding yeast and Mnd1-Hop2 in plants”.

Line 58/59: “.... and its dissociation from dsDNA relies on the Rdh54/Tid1 translocase and its paralog Rad54 in an ATP-dependent manner....” Pl rephrase.

Response: We have revised the sentence as “In yeast, dissociation of Dmc1 from dsDNA relies on the Rdh54/Tid1 ATPase and its paralog Rad54 when in the absence of Rdh54/Tid1...”.

Line 72: “In budding yeast....”: Logic break; why these E3s? Connection sentence missing...

Response: We have added the connection sentence as “Previous studies found that ubiquitination is also important in meiosis”.

Lines 88/89: Check logic; do u really want to say this? Maybe something like this: ...the targets that are controlled by the ub prot pathway are largely uncharacterized.....?

Response: Thank you! We have revised the sentence as “the direct targets under control of the ubiquitin-proteasome system during meiosis are largely uncharacterized”.

Results and Figures:

Suppl. Figure 1B: how was the expression test done? Primers against DMC1 in general or the transgenic part (e.g the FLAG part)?.... the primer set P5 does not indicate this; Please extend the table with a clearer description, indicate in the legend and provide a schematic drawing indicating the positions of the used primer (pl do this for all primer sets and for all (qRT) PCR experiments).

Response: Thanks very much for this suggestion. We have described the qRT-PCR assay in M&M section (Page 21-22 in the revised manuscript) and provided a schematic drawing indicating the positions of the primers used throughout the manuscript (Supplementary Fig. 1b, 8e-g, 10c).

Line 105: “1009 proteins”: pl provide a separate sheet in the file: provide the hits after background removal and indicate those with the GO-term "ubiquitin proteasome pathway", as a service to the reader.

Response: We have added the sheet of the proteins identified by IP-MS analysis of DMC1 after background removal as Supplementary Table 2 and highlighted proteins involved in the GO-term "ubiquitin proteasome pathway" in this sheet as suggested.

Line 111: I could not find the corresponding construct in M&M section and also not the transient transfections with this construct...pl check and complete if needed. This relates to the general aspect mentioned at various occasions: pl make sure that constructs are better described and cross-referenced in the M&M section with respect to the individual experiments.

Response: Thank you! We have added the description of corresponding constructions. Please see the details in “Plasmid construction and plant transformation” in M&M section at Page 18-19.

Suppl. Fig. 2 and 3: in the pptx there is a mis-labelling of the figure # in the right lower corner.

Response: Thanks very much for pointing out the mistake. We have corrected it accordingly.

Line 160: “sterile pollen” does not exist...pl correct.

Response: We have revised the phrase as “inviabile pollen phenotype”.

Line 196: pl tame down sentence:...no proof at that stage of manuscript; rather be specific and say that RMF1 RMF2 act downstream of meiotic DSB formation as does DMC1 and that they do not have an additive effect suggesting that they could....etc..

Response: We appreciate this suggestion and have revised the sentence as “These results provided evidence that RMF1/2 and ASK1 act downstream of meiotic DSB formation as does DMC1 and that they do not have an additive effect, suggesting that they may participate in the same genetic pathway during meiotic recombination”.

Suppl. Figure 5C: Pl. extend figure legend and provide info which heterologous system for expression was used (pl check entire manuscript and clarify all such cases). Moreover, the labelling of the second panel confusing, pl fix.

Response: We have added heterologous expression system in *E.coli* in the revised legend and described the corresponding constructs in detail in M&M section (Page 19). We have also revised the labelling accordingly and checked the entire manuscript to correct such cases.

Figure 4A:

General comment: figure and text flow are not always congruent. Pl fix for entire manuscript (e.g.: Fig 3E/D are discussed after introducing Figure 4A).

Response: Thank you very much for pointing this out! To make the figure and text flow congruence, we have modified the original Fig. 3d-e to the revised Fig. 5a-b, Fig. 4e-h to the revised Supplementary Fig. 6a-d, Fig. 5a to revised Fig. 5c, Fig. 5b to the revised Supplementary Fig. 7a and Fig. 5c to the revised Supplementary Fig. 8a.

Line 233: “to test this hypothesis...”; pl write “to corroborate this hypothesis....”

Response: We have revised the sentence accordingly.

Line 252: sentence confusing: pl rephrase...(you used DMC1 as a substrate, heterologously expressed and purified from *E. coli*.....this is what you mean...right?)

Response: We are sorry for this confusion. Yes, you are right. We have revised the sentence as “we conducted an *in vitro* ubiquitination assay using recombinant DMC1 heterologously expressed and purified from *E. coli* as substrate”.

Line 254: I guess the term “purified” is not appropriate in this context. Use e.g. "captured"

Response: Thank you! We have revised it into “captured” accordingly.

Figure 2:

C-E: Pictures of anthers not as nice as the one in the Suppl. Figure; pl provide better pictures of anthers better demonstrating the full viability.

Response: We have provided better pictures of anthers of Col-0, *rmf1-1* and *rmf2-1* stained with Alexander Red. Please see the revised Fig 2.c-e.

Legend of Figure 2:

Line 870 – “fertile siliques” do not exist...pl rephrase;

Response: We have revised the phrase as “have siliques similar in length to Col-0”.

Line 871 – “sterile siliques” neither...pl rephrase.

Response: We have revised the phrase as “siliques of *rmf1-1 rmf2-1*”.

Line 886 – “have defective” ...pl correct.

Response: We have revised the sentence as “The double mutant of *rmf1-1 rmf2-1* is defective in typical pachytene chromosome and bivalent formation compared with Col-0”.

Line 887 – delete the term “recombination”...this is not assayed here.

Response: We have deleted “recombination” accordingly.

Figure 5C:

Controls are missing: e.g. probing the blot with an anti-Ub antibody; e.g. loading E1 and E2 alone in separate lanes.

The bands that are indicated to represent polyub-DMC1 seem distinct and it is unlikely they represent the 8kDa increments for each single Ub moiety that is additionally attached....

Response: Thanks very much for the great suggestion. We have conducted *in vitro* ubiquitination assay with the control group loading E1 and E2 alone with the substrate, as well as the anti-UBQ11 antibody included. The anti-UBQ11 antibody (Cat. No: AS08 307, Agrisera, Sweden) can recognize both ubiquitin and SUMO, which we also validated (see figure on the right below).

As shown in the new Supplementary Fig. 8a and below, DMC1 can be ubiquitinated in the presence of both E1, E2 with or without SCF^{RMF1/2} as E3. In contrast, DMC1 was not ubiquitinated in the presence of E1, E2 or E3 individually. In addition, using anti-UBQ11 antibody we can also detect the poly-ubiquitinated DMC1. Compared with the anti-DMC1 antibody, the anti-UBQ11 antibody is able to detect the ubiquitin chain on both SUMO-HIS-DMC1 and E2. Since the anti-UBQ11 antibody can also recognize SUMO, the main bands of SUMO-HIS-DMC1 are also recognized by anti-UBQ11 antibody.

a, Ubiquitination of DMC1 *in vitro*.
b, The anti-UBQ11 antibody can recognize ubiquitin and SUMO.

Line 287:

The MS experiment is not described in the M&M section. Pl provide details: endogenous protein; OE lines; tobacco experiments; controls....etc....

Response: We have added the “LC-MS and data processing” in the M&M.

Line 298: the statement that “...suggesting that the mutations do not cause gross structural changes and may not impede recombinase activity ...” has to be deleted as there is no experimental support.

Response: We have deleted this sentence accordingly.

Line 299: please state that it is “nuclear localisation” in tobacco cells....

Response: Thank you! We have revised the sentence as “DMC1 and DMC1-6KR have the same nuclear localization in tobacco cells”.

Figure 5D:

It is not clear which DMC1-OE line was used here...pl specify.

Pl provide entire blot for both ABs....the image is unclear.

Response: Thank you! This assay required large samples, we harvested more than 10 plants from the same transgenic line, and three of them have been validated by western blot and qRT-PCR in the revised Supplementary Fig. 1a-c. We have revised the figure legends to make it clearer. And

we have provided the clear entire blot to replace original one (the revised Fig. 6a).

Figure 5F:

Which of the OE plants (shown and characterized in the supplements...plant 1-3) are used here? Again, pl provide clearer info if cDNA or genomic versions used and if functional complementation tests were performed.

Response: This assay required large samples, we harvested more than 10 plants from the same transgenic line, and three of them have been validated by both WB and qRT-PCR in the revised Supplementary Fig. 8b-j.

We used the cDNA construct as described in the “Plasmid construction and plant transformation” in the M&M section, and have also revised the figure legends to make it clearer.

Figure 6B last lane: double band for DMC1-6KR? Please comment. Furthermore, it appears that the labelling of this lane is not correct. PI check.

Response: We have re-expressed and purified recombinant SUMO-HIS-DMC1 and SUMO-HIS-DMC1-6KR proteins. We found that the original recombinant protein was slightly degraded. We have repeated the *in vitro* pull-down assay (the revised Fig. 7b and see also below) and corrected the labelling.

Validation of the interaction between DMC1-6KR and RMF1/2 by pull-down assay.

Figure 6F: is unclear where the tagged DMC1 protein comes from (expression system, purification).

Response: The tagged DMC1 protein was expressed and purified from *E. coli*. We have added relevant information into the “Recombinant protein expression and purification” in M&M section.

Suppl. Fig. 6 G-J: it is unusual to present new data in the discussion section. Move to the results part.

Moreover, this is most likely one of the most problematic parts of the manuscript. The over-expresser lines are in principle very interesting as they could settle the question if meiotic inter-homologue DNA repair is indeed a fine-tuned aspect of DMC1 protein abundance. Yet the way the experiments have been performed and/or are presented does not allow final conclusions:

Are the presented lines based on cDNA or on genomic DNA....please clarify; It appears that the OE lines are FLAG tagged...pl indicate within the figure. Most importantly: are the over-expressed proteins functional: do they complement the *dmc1* mutant phenotype? This would be crucial, as so far, no tagged and functional version of DMC1 is known in any organism. While the conclusions the authors are drawing could be true (provide all controls would be in place) it could also be that they introduced dominant-negative versions of tagged DMC1, which would then demand a different interpretation. Pl clarify/adapt/explain.

Response: Thanks very much for the reviewer's comments. The transgenic lines used in this figure are based on DMC1 genomic sequence under the control of the *Ler* ecotype *DMC1* promoter or *HTR2* promoter, and FLAG tagged (*pDMC1::DMC1-FLAG* and *pHTR2::DMC1-FLAG*). We have done the complementation assay and found that these constructs do not complement the *dmc1* mutant, and the transgenic lines have similar sterility and meiotic defects compared to *dmc1* mutant. We agree with the reviewer that it is possible that the tagged versions of DMC1 may create unanticipated dominant-negative phenotypes. To avoid drawing improper conclusions, we removed the original Supplementary Figure 6 g-j and corresponding manuscript section in the revised version.

To further validate the function of ubiquitination and protein level of DMC1 *in vivo*, we generated constructs with untagged *DMC1* genomic sequence driven by *RAD51* promoter (*pRAD51::DMC1g*), which restores the meiotic defects of *dmc1* (Da Ines, O. et al. DMC1 attenuates RAD51-mediated recombination in *Arabidopsis*. PLoS Gene, 2022). We introduced both *pRAD51::DMC1g* and *pRAD51::DMC1g-6KR* into *dmc1*. As shown in the revised Supplementary Fig. 10a-e (shown below for convenience), we found that the fertility and meiotic defects were restored much better in *pRAD51::DMC1g/dmc1* transgenic lines. In contrast, all 9 lines of the *pRAD51::DMC1g-6KR/dmc1* showed phenotypes similar to *dmc1*. The qRT-PCR results showed that the transcriptional level of DMC1 was not significantly different between *pRAD51::DMC1g/dmc1* and *pRAD51::DMC1g-6KR/dmc1* transgenic lines. However, the DMC1 protein level was slightly increased in *pRAD51::DMC1g/dmc1* compared with WT (Col-0), and significantly increased in *pRAD51::DMC1g-6KR/dmc1* compared with *pRAD51::DMC1g/dmc1*. These results demonstrated that the K to R mutation might affect DMC1's degradation, and the over-accumulated DMC1 in *pRAD51::DMC1g-6KR/dmc1* could cause meiotic defects similar to *dmc1* mutant. Together, we conclude that of the normal ubiquitination and degradation of DMC1 are critical for its function during meiosis.

- a.** Schematic diagram of *pRAD51::DMC1g* or *pRAD51::DMC1g-6KR* construct.
- b.** Chromosome morphology of Col-0, *dmc1-2*, and two independent lines of *pRAD51::DMC1g/dmc1-2* and *pRAD51::DMC1g-6KR/dmc1-2*, respectively.
- c.** Schematic diagram of DMC1 CDS to show the primer pair used for qRT-PCR in d (DMC1-P1).
- d.** qRT-PCR shows the relative expression level of *DMC1* in inflorescences of Col-0, *dmc1-2*, and two independent lines of *pRAD51::DMC1g/dmc1-2* and *pRAD51::DMC1g-6KR/dmc1-2* with three replicates of individual plants.
- e.** The protein level of DMC1 in Col-0, *dmc1-2*, two independent lines of *pRAD51::DMC1g/dmc1-2* and *pRAD51::DMC1g-6KR*, respectively.

Material and Methods:

Please provide a clearly depiction which constructs make use of genomic versions of a gene, which of cDNA; provide more indicators which construct was used in which experiment (cross-references to Figures/Suppl. Figures). As mentioned at various occasions: make sure to indicate if a construct was tested for functional complementation or not; provide explanations why conclusions may still be valid from given experiment if not tested. Point this out redundantly: in the text body and the M&M section.

Response: We have extensively revised the manuscript and added detailed descriptions of constructs in the M&M section. We have removed the results for the *DMC1-OE* transgenic constructs that failed the functional complementation test (*pDMC1-DMC1g-FLAG/Col-0* and *pHTR2-DMC1g-FLAG/Col-0*, the original Supplementary Fig. 6d-j).

Line 630 ff: details missing regarding elution and further purification; validation of protein ID, purity and yield missing; pl provide.

Response: We have revised the “Recombinant protein expression and purification” in M&M section accordingly.

Discussion:

Line 330: either references or the sentence are wrong; the references relate to RAD51 being ubiquitinated....pl fix.

Response: We checked that the reference (Inano, S. et al. RFW3-Mediated Ubiquitination Promotes Timely Removal of Both RPA and RAD51 from DNA Damage Sites to Facilitate Homologous Recombination. Mol Cell, 2017) we cited here are correct. The reference mainly reported RAD51’s ubiquitination by RING-type E3 RFW3 in human. In addition, the paper also showed that human DMC1 can be modestly ubiquitinated *in vitro* (Figure 4C). Another reference (Rao, H. B. et al. A SUMO-ubiquitin relay recruits proteasomes to chromosome axes to regulate meiotic recombination. Science, 2017) showed that, in mice, when ubiquitination or proteasomes were inhibited, DMC1 foci remained high in late zygotene, consistent with defective turnover (Figure S7), indicating that DMC1 might be under the regulation of ubiquitin-proteasome system.

Lines 355 ff:

“In this study, our higher-order mutants analysis shows that the absence of RAD51 and RMF1/2 and/or ASK1 dramatically decreases meiotic chromosome fragmentation as long as DMC1 is active.”...pl rephrase sentence....it is confusing.

Response: We have revised the sentence as “In this study, our higher-order mutants analysis shows that the absence of RMF1/2 and/or ASK1 dramatically decreases meiotic chromosome fragmentation in *rad51*”.

Line 358: pl use small letters and italics for mutants...otherwise it is confusing.

Response: We have corrected these mistakes accordingly.

Please include the following in your discussion:

Pl discuss why DMC1 was not found in the study for ASK1 targets: Lu et al 2016 PMID: 26940208

Response: Previous proteomics and transcriptomics analyses of ASK1 used floral buds, which include very low percentages of meiocytes, and DMC1 is specifically expressed in distinct stage of meiocytes. This may be why DMC1 was not identified in that study. We would prefer not to include this in the Discussion.

Pl discuss study of PMID: 14756314 and possible redundancy between ASK1 and ASK2

Response: Previous studies showed that, among 21 ASKs in *Arabidopsis*, ASK2 is most similar to ASK1 in terms of sequence similarity, expression pattern, and interaction with the F-box proteins. Genetic analyses found that *ask1* and *ask2* have functional redundancy in multiple developmental processes, but not in meiosis (Liu, F. et al. The ASK1 and ASK2 genes are essential for

Arabidopsis early development. Plant Cell, 2004). Therefore, we speculate that the regulatory mechanism of RMF1/2 targeting DMC1 might be specific association with ASK1 rather than ASK2. We have added this to the first paragraph of our discussion.

Reviewer #2:

The manuscript by Xu et al. described the ubiquitination of a meiosis-specific recombinase, DMC1, by the SCF E3 ubiquitin ligase complex of the SCF with RMF1/2 as an F-box protein in Arabidopsis thaliana. The authors showed that DMC1 binds to RMF-1/2 in both in vivo and in vitro and identified a possible ubiquitination of DMC1 using the expression in tobacco leaf and by in vitro reconstitution. Importantly, the *rmf-1 rmf-2* double mutants and a mutant in the ASK-1 encoding a SKIP1-like protein showed meiotic defects similar to the *dmc1* mutant. In these mutants, the level of DMC1 protein is increased compared to the wild-type control. The same is true for DMC1 protein (DMC1-6KR) with amino-acid substitutions for lysine for possible ubiquitination. This study reveals a new type of regulation of DMC1-mediated meiotic interhomolog recombination, that is associated with ubiquitin-mediated protein degradation. The experiments have been done in a good shape and most of the results are convincing and provide new insight. However, how increased levels of DMC1 in the *ask-1* and *rmf-1 rmf-2* double mutants contribute to meiotic defects is not clearly documented. And the role of ASK1, RMF1/2-dependent removal of DMC1 from meiotic chromosomes has not been analyzed in detail. As shown below, these should be addressed experimentally and discussed well. And the points listed below should be evaluated prior to publication.

Response: We appreciate the reviewer's positive assessment of our manuscript and have carefully considered their comments below.

Major points:

1. In Figures 1 and 6, the authors showed in vivo ubiquitination of DMC1 in tobacco leaves. Interestingly, the main band of both DMC1-GFP or DMC1-FLAG can be detected by not only anti-DMC1 but also anti-UBQ11. Usually, shifted or smear bands of DMC1, not the main band would be reacted with anti-UBQ11. It is strange that DMC1 bands reacted with anti-UBQ11 do not show any upper band shift relative to DMC1 major band detected with either GFP or DMC1 antibodies. Does this mean that the major DMC1 band is a constitutively mono-ubiquitinated DMC1? In the same line, TUBE2-purified DMC1, which should be ubiquitinated, also shows a sharp band in Figures 1D, E, and 6C, suggesting that the major DMC1 band contains the ubiquitin-binding ability reacted with TUBE2. It is essential to explain these strange results.

Response: This is a great question. To verify whether the major DMC1 band is a constitutively mono-ubiquitinated DMC1, we purified recombinant SUMO-HIS-DMC1-GFP and SUMO-HIS-DMC1-FLAG from *E.coli*, then the SUMO-HIS tag was cleaved by SUMOase to yield DMC1-GFP and DMC1-FLAG. Because DMC1 is not ubiquitinated in *E.coli*, the molecular weight of DMC1-GFP from *E.coli* is smaller than the expressed DMC1-GFP in tobacco and DMC1-FLAG from Arabidopsis. Moreover, anti-UBQ11 antibody (Cat. No: AS08 307, Agrisera, Sweden) can recognize both ubiquitin and SUMO, which we validated in our experiments (see figure in page 6 response to Reviewer #1). Therefore, anti-UBQ11 antibody can recognize both DMC1-GFP and DMC1-FLAG expressed in tobacco and DMC1-FLAG expressed in Arabidopsis

after immunoprecipitated by using anti-GFP or FLAG antibodies, but does not recognize DMC1-GFP and DMC1-FLAG from *E.coli* (revised Supplementary Fig. 1e, f). Taken together, our results demonstrated that DMC1 can be mono- and poly-ubiquitinated *in planta*, in which the main band might be a constitutively mono-ubiquitinated DMC1. This paper is focused on the DMC1 degradation, which is likely caused by poly-ubiquitination, while the biological role of mono-ubiquitinated DMC1 is deserved for further study.

Validation of DMC1's mono-ubiquitination.

2. The univalent formation in the *ask-1*, *rmf-1 rmf-2*, *ask-1 rad51* and *rmf-1 rmf-2 rad51* mutants (Figure 3A, D) suggests the DSBs are repaired by inter-sister recombination. Thus, these mutants with wild-type or more DMC1 focus number (Figures 5A and B) can dismantle DMC1 from the chromosomes for the completion of the recombination without ASK1 or RMF1/2. On the other hand, the authors claim the role of ASK1-RMF1/2 removal of DMC1 in wild-type situations, but not in the mutants (Figure 7). This discrepancy should be discussed well. In the line, it is very important to check the DMC1 focus formation (with the quantification) in the late stages of meiotic prophase I such as diakinesis in the *ask-1*, *rmf-1 rmf-2*, *ask-1 rad51*, and *rmf-1 rmf-2 rad51* mutants. If ASK1-RMF1/2 can remove DMC1 filaments for proper interhomolog recombination (Figure 7), one can expect the accumulation of DMC1 foci on chromosomes in diakinesis.

Response: As suggested by the reviewer, we conducted immunofluorescence staining using anti-DMC1 antibody to investigate DMC1 foci in diakinesis meiocytes, but we did not observe any DMC1 signal in Col-0, *ask1*, *rmf-1 rmf2*, *ask1 rad51*, or *rmf1 rmf2 rad51* mutants (See the figures below). We did not include this experiment in the revised manuscript. Instead, we have removed the DMC1 from single end DSB during the step of strand displacement in the revised model (the revised Fig. 8).

DMC1 foci localization of Col-0, *ask1-1*, *rmf1-1 rmf2-1*, *ask1-1 rad51-1* and *rmf1-1 rmf2-1 rad51-1* at diakinesis.

3. The univalent formation in the *ask-1 rad51* and *rmf-1 rmf-2 rad51* mutants also suggests that, in

these mutants, DMC1 cannot carry out interhomolog recombination, but rather can do intersister recombination. This point should be discussed. And, importantly, to extend the idea, the chromosome synapsis should be studied by the staining for ZYP1A in these mutants.

Response: Thanks for these suggestions. We have examined chromosome synapsis by immunofluorescence staining for ASY1 and ZYP1A in *ask1 rad51*, *rmf1 rmf2 rad51* and *ask1 rmf1 rmf2 rad51* mutants. We did not observe any linear ZYP1A signals in the mutants compared with WT (revised Supplementary Fig. 4c and below, right 3 panels), suggesting a synapsis defect. It is possible that DMC1 is active in these mutants and may substitute for RAD51 during inter-sister recombination.

SYN1 (red) and ZYP1A (green) distribution of Col-0, *ask1-1*, *rmf1-1 rmf2-1*, *dmc1-2* and *ask1-1 rad51-1*, *rmf1-1 rmf2-1 rad51-1*, *ask1-1 rmf1-1 rmf2-1 rad51-1* at pachytene.

4. Figure 6, the treatment with MG132 stabilizes the steady-state level of wild-type DMC1 in tobacco leaves (increased 10-fold or ~4-fold in Figure 5E). In the tobacco leaf system, what happens to the expression of RMF1/2 and ASK1? Indeed, does the ubiquitination and/or degradation of DMC1 depend on both RMF1/2 and ASK1?

Response: This is a good question. Our original results demonstrated that expression of RMF1/2 in tobacco can promote DMC1's degradation (revised Fig 6d), and DMC1 protein levels are also significantly higher in both *rmf1 rmf2* and *ask1* mutants compared with Col-0, indicating that the ubiquitination and degradation of DMC1 depend on both RMF1/2 and ASK1. To further verify whether DMC1's degradation is also dependent on ASK1, *in vivo* degradation assay was performed in transient expression in tobacco. With increasing amounts of ASK1, the protein level of DMC1 decreased gradually, indicating that ASK1 is also responsible for DMC1's ubiquitination and degradation (Revised Fig. 6e and also see below). We have also tried to simultaneous expression RMF1/2, ASK1 and DMC1 in tobacco several times, but were not successful.

ASK1 promotes DMC1 degradation in a tobacco transient expression system.

5. The suppression of the *rad51* mutant by the SCF mutants is similar to the results in the recent publication showing that a mutation in the SMC5/6 genes suppresses the *rad51*'s meiotic defects including DMC1 focus formation (Chen, Plant Cell, 33, 2869-, 2021). The relationship of ASK1-RMF1/2 with SMC5/6 should be experimentally addressed or discussed.

Response: This is a good question. Although both SMC5/6 and ASK1-RMF1/2 have a similar effect on suppressing the *rad51*'s meiotic chromosome fragments, the underlying mechanisms are different between them. SMC5/6 can rescue the *rad51* partial fertility, homolog synapsis and chromosome fragments (Chen, H. et al. RAD51 supports DMC1 by inhibiting the SMC5/6 complex during meiosis. Plant Cell, 2021), but the ASK1-RMF1/2 only rescues the chromosome fragments much better than that of SMC5/6, but no effect on fertility and synapsis, suggesting that, in the absence of RAD51, SMC5/6 and ASK1-RMF1/2 may have different roles in mediated-DMC1 dependent inter-homolog and inter-sister recombination. It is also plausible that SCF^{RMF1/2} functions downstream of SMC5/6. However, these hypotheses deserve further study and are beyond the scope of this manuscript.

Minor points:

1. Figure 4E and 4F: These results should be in Supplement without the analysis of the domains for *in vivo* or *in vitro* ubiquitination.

Response: We have moved the corresponding figures to supplementary figures accordingly.

2. Figures 5A and 5B: Please add DMC1 focus counting in pachytene or pachytene-like stages in each strain to validate the claim (lines 236-238).

Response: Thank you. We have added DMC1 foci counting in pachytene or pachytene-like stages in revised Supplementary Fig. 7a.

3. Figure 5D bottom panel: Please show a full blot with a DMC1-FLAG band like in Figures 1D and 1E.

Response: We now show the full blot in revised Fig. 6a (see also below).

RMF1/2 are responsible for DMC1 ubiquitination *in vivo*.

4. Lines 179-181: *ask-1 spo11* and *rmf-1 rmf-2 spo11* are clearly different from *spo11* in anaphase I chromosome morphology. The *spo11* shows a more compacted univalent while the *ask-1 spo11* and *rmf-1 rmf-2 spo11* mutants show an elongated or V-shaped univalent. This suggests that the *ask-1* and *rmf-1 rmf-2* mutants show Spo11-independent meiotic defects (opposed to the claim

shown in lines 182-184). This point should be clarified, and the authors need a quantitative analysis of chromosome defects.

Response: This is a great observation. We agree with the reviewers that, in addition to meiotic recombination, RMF1/2 and ASK1 have other roles during meiosis, such as sister chromosome cohesion as suggested by the V-shaped univalent, but this function could be independent of DMC1, and is not the focus of this study. To avoid confusion, we revised the text to focus the comparison of the meiotic phenotype on univalent or recombination.

5. Line 195, meiotic recombination defects: Figure 3C shows only chromosome defects, not recombination defects. The authors need to be careful about the conclusion from the results.

Response: We have revised the “meiotic recombination defects” into “meiotic chromosome morphology defects”.

6. Line 199: The authors pointed out ASK2 in addition to ASK1. Since *ask-1* single mutant showed meiotic defects. The authors need more explanation on the ASK2 more in its function.

Response: Reviewer #1 also raised this question. Although ASK1 and ASK2 have high similarity or sequence similarity, they have distinct roles in growth and development. ASK1 but not ASK2 is required for meiosis, while both ASK1 and ASK2 are redundantly required for plant development rather than meiosis (Liu, F. et al. The ASK1 and ASK2 genes are essential for Arabidopsis early development. *Plant Cell*, 2004). Given that RMF1/2 only functions in meiosis, the *in vitro* interaction between RMF1-ASK2 might not reflect actual function *in vivo*.

7. Line 268: What is the reason why the authors used Mg²⁺ as a control of MG-132?

Response: The suspension buffer we used including 10 mM 4-morpholineethanesulfonic acid (MES), 100 μM acetosyringone (AS), and 10 mM MgCl₂, of which MES and AS were used to promote the expression of heterologous protein in tobacco leaves. To maintain the Mg²⁺ balance, we used 10 mM MgCl₂ as control referred to previous study (Liu, L. et al. An efficient system to detect protein ubiquitination by agroinfiltration in *Nicotiana benthamiana*. *Plant J*, 2010).

8. Line 271-273: What is the phenotype of RMF1/2 or ASK1 OE plant? Given the reduced DMC1 levels, these mutants should show a meiotic recombination defect like the *dmc1*. DMC1 focus and chromosomal abnormality should be presented.

Response: This is a good question! Theoretically, the overexpression of either RMF1/2 or ASK1 will cause a similar meiotic defect similar to *dmc1*, due to the reduction of DMC1 as pointed out by the Reviewer. Unexpectedly, we did not observe any defects in fertility and meiosis of the *RMF1/2* or *ASK1-OE* plants. We thought that, as shown in revised Fig. 6c, the DMC1 level in *RMF1/2* or *ASK1-OE* are sufficient for successful meiosis and fertility.

Reviewer #3 (Remarks to the Author):

The manuscript “SCF^{RMF}-mediated degradation of the recombinase DMC1 ensures meiotic recombination” by Wang and co-authors presents the results of an in-depth study of Arabidopsis

DMC1 and more specifically how its regulated degradation is essential for homologous recombination during meiosis. The authors describe an elegant series of experiments that result in the identification of the redundant F-box genes RMF1/2 and show that RMF1/2 interact with the Skp1 ortholog ASK1 to form a ubiquitin ligase complex SCF^{RMF1/2} that is responsible for DMC1 ubiquitination and its subsequent degradation by the 26S proteasome.

The results of the work are highly original further our understanding of how DMC1 is regulated and make important contributions to the field of meiotic homologous recombination, and meiosis in general. This is the first evidence that DMC1 is ubiquitinated in plants and I believe the first instance of results describing a detailed mechanistic picture of how specifically DMC1, a key meiotic recombinase that is broadly conserved in eukaryotes is regulated in any organism.

The manuscript is generally well-written, and the data presented is extensive and of high quality. The conclusions drawn are significant and are supported by the result presented. In many instances the authors present the results of multiple experiments in support of a particular conclusion. My only suggestion is that the manuscript be given a careful proof-reading as there are many small grammatical errors sprinkled throughout the manuscript.

Response: We appreciate the reviewers' positive comments for our work. We have extensively revised the manuscript by a native English speaker.

REVIEWER COMMENTS

Reviewer #2 (Remarks to the Author):

NCOMMS-22-45529A

The authors responded to my previous comments in a proper way. But some additional results generate new comments on the results in this paper. Mainly, a new result clearly showed the mono-ubiquitination of most of the Dmc1 protein in plant cells. This modified version of the Dmc1 protein looks quite stable in a cell, suggesting that the mono-ubiquitination is nothing to do with the ubiquitin-mediated protein degradation by the proteasome. So the authors need to clarify multi-ubiquitination and mono-ubiquitination in the text. In the current version, the authors mainly described the role of multi-ubiquitination of Dmc1 by a novel F-box protein RMI1/2. However, the interpretation of results using a non-ubiquitinated mutant version of Dmc1, such as Dmc1-6KR, could be re-interpreted based on both forms of ubiquitination. Moreover, the fact that Dmc1-6KR can not complement the *dmc1* defect raises doubt about whether RMI1/2-mediated ubiquitination (degradation) is physiologically relevant or not; e.g. The down-regulation of the Dmc1 protein level is critical for meiotic recombination. Alternatively, there is the other target RMI1/2-mediated ubiquitination (degradation), which is essential for Dmc1-mediated recombination.

The following point should be clearly checked.

1. It is critical to check whether mono-ubiquitinated Dmc1 is indeed present in the meiocytes of plants (or induced by somatic expression) since the authors showed the mono-ubiquitination only in the tobacco cells and/or transgenic Arabidopsis with mitotically-expressed Dmc1 (Figure 1b-e). This can be done by comparing endogenous Dmc1 in meiocytes with Dmc1 expressed in *E. coli* (used in Supplemental Figures 1e and f) by simple western blotting analysis (as shown in Figure 6c). Alternatively, by probing fractions from TUBE2 bead mixed with lysates from meiocytes by the anti-Dmc1 antibody in non-transgenic wild-type (not Anti-Flag shown using a transgenic line with Dmc1-Flag in Fig. 6A).
2. Mono-ubiquitinated Dmc1 is not dependent on RMI1/2 or ASK1 (Figure 6a). Dmc1-6KR is defective in mono-ubiquitination (Figure 7). Re-interpretation of mutant defects in Dmc1-6KR transgenic lines is necessary based on the defective mono-ubiquitination. Figure 8 should be revised by adding the mono-ubiquitinated Dmc1 (if the mono-ubiquitinated form of Dmc1 is present in meiotic cells).
3. The result of the new experiment (Supplemental Figure 10e) showed that the size of wild-type Dmc1 is similar to that of Dmc1-6KR, which showed defective mono-ubiquitination (Figure 7c). This is very strange if most of Dmc1 is mono-ubiquitinated. The authors need to explain it or experimentally address it. Again, this raises the possibility that ubiquitination, at least the mono-ubiquitination, of Dmc1 is not critical for the *in vivo* function. The authors can check the role of the Dmc1 ubiquitination by checking the suppression of Dmc1-6KR mutation in *rad51* background as shown in *rmi1/2* or *ask1* mutations as shown in Figure 5 (by relieving entanglement defect or increased Dmc1 foci).
4. Please add the quantification of the focus number of Dmc1 in Figure 5c.

Point-by-point response to reviewer's comments

Thank you very much for your helpful comments and suggestions for our manuscript entitled “SCF^{RMF} Mediates Degradation of the Meiosis-Specific Recombinase DMC1” (Tracking NO. 22-45529-A). We are pleased to resubmit our revised manuscript for your consideration. Below, please find our point-by-point response (questions in black and responses in blue).

Reviewer #2 (Remarks to the Author):

NCOMMS-22-45529A

The authors responded to my previous comments in a proper way. But some additional results generate new comments on the results in this paper. Mainly, a new result clearly showed the mono-ubiquitination of most of the Dmc1 protein in plant cells. This modified version of the Dmc1 protein looks quite stable in a cell, suggesting that the mono-ubiquitination is nothing to do with the ubiquitin-mediated protein degradation by the proteasome. So the authors need to clarify multi-ubiquitination and mono-ubiquitination in the text. In the current version, the authors mainly described the role of multi-ubiquitination of Dmc1 by a novel F-box protein RMI1/2. However, the interpretation of results using a non-ubiquitinated mutant version of Dmc1, such as Dmc1-6KR, could be re-interpreted based on both forms of ubiquitination. Moreover, the fact that Dmc1-6KR can not complement the *dmc1* defect raises doubt about whether RMI1/2-mediated ubiquitination (degradation) is physiologically relevant or not; e.g. The down-regulation of the Dmc1 protein level is critical for meiotic recombination. Alternatively, there is the other target RMI1/2-mediated ubiquitination (degradation), which is essential for Dmc1-mediated recombination.

Response: We appreciate the reviewer's positive assessment of our manuscript and have carefully considered those comments and revised the manuscript accordingly.

The following point should be clearly checked.

1. It is critical to check whether mono-ubiquitinated Dmc1 is indeed present in the meiocytes of plants (or induced by somatic expression) since the authors showed the mono-ubiquitination only in the tobacco cells and/or transgenic *Arabidopsis* with mitotically-expressed Dmc1 (Figure 1b-e). This can be done by comparing endogenous Dmc1 in meiocytes with Dmc1 expressed in *E. coli* (used in Supplemental Figures 1e and f) by simple western blotting analysis (as shown in Figure 6c). Alternatively, by probing fractions from TUBE2 bead mixed with lysates from meiocytes by the anti-Dmc1 antibody in non-transgenic wild-type (not Anti-Flag shown using a transgenic line with Dmc1-Flag in Fig. 6A).

Response: Thanks for these suggestions. As suggested, we purified recombinant SUMO-HIS-DMC1 from *E. coli* and then used SUMOase to cleave the SUMO-HIS tag to yield untagged DMC1. Since it is not feasible to obtain sufficient meiocytes to do western blots, we instead collected *Arabidopsis* Col-0 central inflorescences to examine the endogenous DMC1. *Arabidopsis* peripheral inflorescences and leaves, and *dmc1-2* central inflorescences were used as

controls. We examined DMC1 expressed in *E.coli* and endogenous *Arabidopsis* DMC1 by western blot using an anti-DMC1 antibody, the result show that DMC1 expressed in *E.coli* exhibits the same molecular weight as the endogenous DMC1 in *Arabidopsis* central inflorescences, and that DMC1 cannot be detected in Col-0 peripheral inflorescences and leaves, or in *dmc1-2* central inflorescences (See Supplementary Fig. 1g and the figure below). This result indicates that the mono-ubiquitinated DMC1 is not present at detectable levels in meiocytes. We appreciate the reviewer pointing out this and agree that mono-ubiquitinated DMC1 could be induced by ectopic expression in somatic cells. We have added these results as Supplementary Fig. 1g and rephrased the corresponding text (lines 140-151) in the revised manuscript.

2. Mono-ubiquitinated Dmc1 is not dependent on RMI1/2 or ASK1 (Figure 6a). Dmc1-6KR is defective in mono-ubiquitination (Figure 7). Re-interpretation of mutant defects in Dmc1-6KR transgenic lines is necessary based on the defective mono-ubiquitination. Figure 8 should be revised by adding the mono-ubiquitinated Dmc1 (if the mono-ubiquitinated form of Dmc1 is present in meiotic cells).

Response: This is related to question #1, please see our response above. Mono-ubiquitinated DMC1 is only induced by somatic expression, but the six lysine residues in DMC1 identified by mass spectrometry can be mono- and poly-ubiquitinated. Therefore, both mono- and poly-ubiquitination level of DMC1-6KR expressed in tobacco leaves were defective. Since DMC1 is not present at detectable levels in meiotic cells we did not add the mono-ubiquitinated form of DMC1 to the model.

3. The result of the new experiment (Supplemental Figure 10e) showed that the size of wild-type Dmc1 is similar to that of Dmc1-6KR, which showed defective mono-ubiquitination (Figure 7c). This is very strange if most of Dmc1 is mono-ubiquitinated. The authors need to explain it or experimentally address it. Again, this raises the possibility that ubiquitination, at least the mono-ubiquitination, of Dmc1 is not critical for the in vivo function. The authors can check the role of the Dmc1 ubiquitination by checking the suppression of Dmc1-6KR mutation in *rad51* background as shown in *rmi1/2* or *ask1* mutations as shown in Figure 5 (by relieving entanglement defect or increased Dmc1 foci).

Response: This question is also related to both questions above. We have now demonstrated that the mono-ubiquitinated DMC1 was induced by ectopic expression in somatic cells. We agree with the reviewer it would be interesting to test the DMC1-6KR mutation in *rad51* background, but it would take at least half of year to obtain the necessary transgenic plants. Furthermore, since mono-ubiquitinated DMC1 is not present at detectable levels in meiocytes and DMC1-6KR fails to complement the *dmc1* mutant, we speculate that DMC1-6KR is unlikely to relieve the *rad51* meiotic recombination defects. We hope its satisfactory for the reviewer.

4. Please add the quantification of the focus number of Dmc1 in Figure 5c.

Response: We have provided the requested quantification in Supplementary Fig. 7a.

REVIEWERS' COMMENTS

Reviewer #2 (Remarks to the Author):

The authors addressed my major concern about the mono-ubiquitination of Dmc1 in the in vivo expression system. Their result showed that Dmc1 mono-ubiquitination was not observed in meiocytes. This result casts doubt on the implication of the results on Dmc1-ubiquitination detected by the in vivo system.

The results in this paper clearly show the RMF1/2-mediated degradation of Dmc1. On the other hand, the authors do not address how meiocytes maintain a level of Dmc1, which escapes RMF1/2-mediated degradation, to ensure meiotic interhomolog recombination. Alternatively, the authors do not address how a 2-fold excess level of Dmc1 in the *rmf1/2* mutant or the *ask1* mutant blocks Dmc1-mediated recombination.

Point-by-point response to reviewer's comments

Thank you very much for your helpful comments and suggestions for our manuscript entitled “SCF^{RMF} Mediates Degradation of the Meiosis-Specific Recombinase DMC1” (Tracking NO. NCOMMS-22-45529B). We are pleased to resubmit our revised manuscript for your consideration. Below, please find our point-by-point response (questions in black and responses in blue).

Reviewer #2 (Remarks to the Author):

The authors addressed my major concern about the mono-ubiquitination of Dmc1 in the *in vivo* expression system. Their result showed that Dmc1 mono-ubiquitination was not observed in meiocytes. This result casts doubt on the implication of the results on Dmc1-ubiquitination detected by the *in vivo* system.

Response: We appreciate the reviewer's satisfaction to our response. Our *in vivo* and *in vitro* ubiquitination assay provided strong evidence that DMC1 can be poly-ubiquitinated. We speculate that the poly-ubiquitination-dependent degradation of DMC1 is important to maintain DMC1 homeostasis, thus ensuring meiotic recombination.

The results in this paper clearly show the RMF1/2-mediated degradation of Dmc1. On the other hand, the authors do not address how meiocytes maintain a level of Dmc1, which escapes RMF1/2-mediated degradation, to ensure meiotic interhomolog recombination. Alternatively, the authors do not address how a 2-fold excess level of Dmc1 in the *rmf1/2* mutant or the *ask1* mutant blocks Dmc1-mediated recombination.

Response: This is a good question! As responded above, the RMF1/2-ASK1 module is required for maintaining DMC1 homeostasis rather than complete degradation of DMC1. We hypothesize that over DMC1 level has a role in blocking DMC1-mediated recombination likely via feedback effect on the removal of DMC1 from single end DNA-protein filament. As suggested by the editor, we have added the sentence “However, RMF1/2-mediated degradation of DMC1 maintaining its proper level to ensure meiotic inter-homolog recombination needs further investigation” at the end of discussion section. We hope that the reviewer will satisfy to this explanation.